



Earth System
Dynamics

# Tidal impacts on primary production in the North Sea

**Changjin Zhao, Ute Daewel, and Corinna Schrum**

Helmholtz Centre Geesthacht, Institute of Coastal Research, Max-Planck-Str. 1, 21502 Geesthacht, Germany

**Correspondence:** Changjin Zhao (changjin.zhao@hzg.de)

**Abstract.** This study highlights the importance of tides in controlling the spatial and temporal distributions of phytoplankton and other factors related to growth, such as nutrients and light availability. To quantify the responses of net primary production (NPP) to tidal forcing, we conducted scenario model simulations considering $M_2$ and $S_2$ tidal constituents using the physical–biogeochemical coupled model ECOSMO (ECOSystem MOdel). The results were analyzed with respect to a reference simulation without tidal forcing, with particular focus on the spatial scale of the tidally induced changes. Tidal forcing regulates the mixing–stratification processes in shelf seas such as the North Sea and hence also influences ecosystem dynamics. In principle, the results suggest three different response types with respect to primary production: (i) in southern shallow areas with strong tidal energy dissipation, tidal mixing dilutes phytoplankton concentrations in the upper water layers and thereby decreases NPP. Additionally, tides increase turbidity in near-coastal shallow areas, which has the potential to further hamper NPP. (ii) In the frontal region of the southern North Sea, which is a transition zone between stratified and mixed areas, tidal mixing infuses nutrients into the surface mixed layer and resolves summer nutrient depletion, thus sustaining the NPP during the summer season after spring bloom nutrient depletion. (iii) In the northern North Sea, the NPP response to tidal forcing is limited. Additionally, our simulations indicate that spring bloom phenology is impacted by tidal forcing, leading to a later onset of the spring bloom in large parts of the North Sea and to generally higher spring bloom peak phytoplankton biomasses. By testing the related changes in stratification, light conditions and grazing pressure, we found that all three factors potentially contribute to the change in spring bloom phenology with clear local differences. Finally, we also analyzed the impact of the spring–neap tidal cycle on NPP. The annual mean impact of spring–neap tidal forcing on NPP is limited. However, locally, we found substantial differences in NPP either in phase or anti-phase with the spring–neap tidal cycle. These differences could be attributed to locally different dominant factors such as light or nutrient availability during spring tides. In general, we conclude that in shallow shelf seas such as the North Sea, intensified vertical mixing induced by tidal forcing could either promote NPP by counteracting nutrient depletion or hinder NPP by deteriorating the light environment because of the resuspension and mixing of suspended matter into the euphotic zone.

## 1   Introduction

Coastal and shelf seas, such as the North Sea, generally show primary production up to 3–5 times that of the open ocean (Simpson and Sharples, 2012). Among the potential reasons for this difference are the tides, one of the dominant physical forcing factors in the North Sea, which regulate the mixing–stratification status (Pingree and Griffiths, 1978; Simpson and Souza, 1995), with potential implications for primary production (Daly and Smith, 1993; Otto et al., 1990). The relevance of tides to primary production has been investigated in a number of previous studies, which show substantial co-variability between hydrodynamic tidal characteristics and biogeochemical data (Blauw et al., 2012; Jago et al., 2002; McCandliss et al., 2002; Pietrzak et al., 2011; Richardson et al., 2000). Tides influence biogeochemical cycling in various ways, enhancing the vertical mixing of biomass, suspended matter and nutrients, and causing sediment resuspen-

sion. Vertical mixing injects nutrients (e.g., Hu et al., 2008) into the euphotic zone and thereby sustains primary production. However, vertical mixing also promotes the dilution of phytoplankton biomass (Cloern, 1991), which hinders plankton production. The resuspension and upward vertical mixing of near-bottom sediments (Bowers et al., 1998; Smith and Jones, 2015) deteriorate light conditions (Porter et al., 2010) and result in decreasing productivity. The co-action of these mechanisms results in either favorable or unfavorable impacts on ecosystem productivity depending on local hydrodynamic and biochemical conditions, thus shaping the specific structure and sensitivity of North Sea net primary production (NPP).

In the North Sea, several subsystems emerge with respect to tidal forcing and bathymetry, leading to a high spatial diversity of primary production dynamics (Van Leeuwen et al., 2015) and potentially also NPP sensitivity to tides. In principle, the system can be differentiated into a permanently mixed shallow area in the southern North Sea, a seasonally stratified area in the central and northern North Sea and a transition zone that includes frontal and weakly stratified areas (Schrum et al., 2003). In permanently mixed shallow areas, strong vertical stirring slows the development of the spring bloom and prevents summer nutrient limitation (Wafar et al., 1983). Nutrient availability in shallow coastal areas is additionally enhanced by onshore nutrient and organic matter transport driven by estuarine-type baroclinic circulation (Hofmeister et al., 2017; Rodhe et al., 2004) and land-borne nutrient supplies. Consequently, light limitation is dominant in shallow coastal areas (Tett and Walne, 1995). In contrast, the central and deeper parts of the northern North Sea are seasonally stratified (Pohlmann, 1996), and summer nutrient depletion occurs in the upper mixed layer after the spring bloom (Longhurst, 2006). Because the bottom mixed and surface mixed layers in these regions are largely decoupled, the tidally driven nutrient replenishment from the deeper layers is expected to be rather small. In shallower areas, the bottom mixed layer is able to interfere with the thermocline, and nutrients can be mixed into the euphotic zone (e.g., Rippeth et al., 2009; Richardson et al., 2000; Sharples, 2008; Daewel and Schrum, 2013) and sustain the NPP in the euphotic zone in summer. In these areas, the breaking up of stratification is mainly driven by the spring–neap tidal cycle or wind mixing (Mahadevan et al., 2010; Schrum, 1997). The physical mechanisms of the spring–neap cycle, such as the shifting of fronts (Simpson and Bowers, 1981), periodical erosion of the thermocline and relevant ecological responses (Allen et al., 2004), mainly in regard to replenishment of nutrients (Franks and Chen, 1996) and interruption of biomass building (Balch, 1981; Sharples et al., 2006), have been studied previously. In addition to large-scale stratification patterns that regulate tidal impacts on NPP, local impacts have been observed. The patchiness of chlorophyll (CHL) concentrations at the eastern British coast, for example, was shown to be associated with local vertical mixing generated by tides and bathymetry

(Scott et al., 2010). In the Rhine river plume area, suspended particulate matter concentrations are characterized by a periodicity following a fortnight cycle (Pietrzak et al., 2011).

So far, earlier studies have focused largely on the local effects of nutrient injection into the euphotic zone. Understanding key processes and assessing regionally differing responses have been accomplished by cross-frontal field studies and idealized model simulations (e.g., Cloern, 1991; Richardson et al., 2000; Sharples, 2008). Some of these studies have quantitatively evaluated tidal contributions to NPP based on nutrient replenishment from observed data or 1-D simulations using simplified upscaling, neglecting the spatial diversity of the North Sea system. However, it remains an open question how dynamic zooplankton and tide-modulated benthic–pelagic coupling affect the sensitivity of plankton production to tidal forcing. Furthermore, a comprehensive understanding of tidal impacts at a basin scale is still lacking for the North Sea. To answer these questions and investigate highly dynamic tidal impacts on ecosystem productivity in different subsystems in the North Sea, the application of 3-D modeling is indispensable. Here, we will address the above questions using ECOSMO (ECOSystem MOdel) (Daewel and Schrum, 2013; Schrum et al., 2006), a well-validated 3-D-coupled physical–biogeochemical model for scenario simulations to elaborate the relevance of tidal impacts on NPP and underlying processes. The model resolves key physical and biogeochemical processes, such as turbulent mixing, zooplankton growth and predation, and impacts of particulate and dissolved organic matter on light conditions. The model has a bottom component, which is dynamically coupled to the water column through the fluxes of particulate and dissolved matter, allowing for resuspension. We will assess the spatial variability of the responses of NPP to major tidal components, i.e., $M_2$ and $S_2$, and disentangle different processes contributing to tidally induced variations in NPP, mainly variations related to stratification–mixing patterns, spring bloom onset time and intensity, and the maintenance of NPP in the subsurface of stratified areas. We will further investigate variations in NPP related to the spring–neap tidal cycle.

## 2 Methods

### 2.1 Model description and validation

In this study, we employed the well-validated 3-D-coupled physical–biochemical model ECOSMO (Daewel and Schrum, 2013). The hydrodynamic component of ECOSMO builds on the 3-D baroclinic model HAMSOM (HAMburg Shelf Ocean Model) (Schrum and Backhaus, 1999). The capability to simulate the hydrodynamic status of the North Sea–Baltic Sea system was validated by Janssen et al. (2001) and Schrum et al. (2003). The simulation domain covers the North Sea and Baltic Sea, with open boundaries to the northern Atlantic Ocean in the north and the mouth

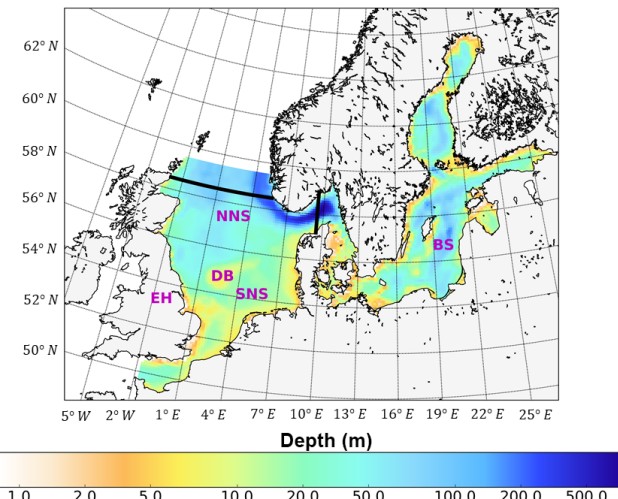

**Figure 1.** Bathymetry and simulation domain of ECOSMO. Black lines indicate the area of the North Sea used for analysis, from 5° W to 9.5° E in the east–west direction and from 48 to 58.5° N in the south–north direction. SNS and NNS are short for the southern and northern North Sea, respectively. BS is short for the Baltic Sea. DB and EH are short for Dogger Bank and the Estuary of Humber, respectively.

of the English Channel in the south (Fig. 1). The model was formulated on a staggered Arakawa-C grid using spherical coordinates, with a spatial resolution of 6′ in latitude and 10′ in longitude. The model time step was 20 min, which allows for a robust representation of the tidal cycle for physics and biogeochemistry. It was also coupled online using the same time steps as those for hydrodynamics. In this study, we focused on the North Sea region between 48–58.5° N and 5° W–9.5° E because tides are only of minor relevance in the Baltic Sea. To resolve thermal stratification in the upper water column, the vertical resolution was set to 5 m in the upper 40 m of the water column and decreased gradually with depth below 40 m. To reduce numerical diffusion in the implemented upwind advection scheme, a shape-preserving total-variation-diminishing (TVD) scheme (Yee et al., 1985) was adopted, which significantly improved the representation of hydrodynamics and ecosystem processes, especially processes related to fronts. A detailed description of the method and model responses to the changed advection scheme has been provided by Barthel et al. (2012).

The biogeochemical component of ECOSMO was developed to describe the lower trophic level dynamics of the marine ecosystem using a nutrient–phytoplankton–zooplankton–detritus (NPZD) conceptual model framework. The ecosystem model component was first introduced for the North Sea by Schrum et al. (2006) and further developed for a wider range of ecosystems, including relevant characteristics for the Baltic Sea, by Daewel and Schrum (2013). Detailed validations against nutrient observations have shown that the model is capable of simulating lower trophic level ecosys-

tem dynamics in the North Sea, and the temporal variability at interannual to decadal scales simulated by ECOSMO could be corroborated by observations (Daewel and Schrum, 2013). ECOSMO simulates the nutrient cycling of silicate, phosphorus and nitrogen in the water column and in the sediments considering processes such as primary production, grazing and excretion by zooplankton, remineralization and sediment–water coupling. A detailed description of the ecosystem model is given in Daewel and Schrum (2013). In total, 16 state variables were solved, including three functional groups for primary producers (diatoms, flagellates and cyanobacteria). In the second trophic level, two groups of zooplankton were considered and differentiated based on feeding preferences. To additionally account for the shading effects of dissolved organic matter (DOM) and detritus, which were not considered in Daewel and Schrum (2013), the formulation of light attenuation was modified as previously suggested by Nissen (2014). To capture the productive and turbid characteristics, DOM was parameterized by fast remineralization rates and a low sinking velocity, in contrast to the fast sinking velocity and slow remineralization rates of particulate organic matter (detritus). Therefore, the vertical light attenuation consisted of background attenuation ($k_{w1}$) (induced by the water body and inorganic SPM), phytoplankton self-shading ($k_p$) and additional shading impacts of DOM ($k_{DOM}$) and detritus ($k_{Det}$), as shown in Eq. (1).

$$Kd_1 = k_{w1} + k_p \cdot P + k_{DOM} \cdot DOM + k_{Det} \cdot Det \qquad (1)$$

While background attenuation $k_{w1}$ (0.03 m$^{-1}$; Urtizberea et al., 2013) remained constant in the water column, self-shading depended on both $k_p$ (0.2 m$^2$ mmol C$^{-1}$) and the phytoplankton concentration ($P$). As suggested by Stedmon et al. (2000) and Tian et al. (2009), $k_{DOM}$ and detritus $k_{Det}$ were set to 0.29 m$^2$ gC$^{-1}$ and 0.2 m$^2$ gC$^{-1}$, respectively. Compared to Daewel and Schrum (2013), these changes enabled the dynamical coupling of turbidity to the seasonal production cycle, as previously discussed by Nissen (2014). A corresponding validation of surface nutrients and comparison of mean primary production (Appendix B) confirms that the performance of ECOSMO in the North Sea region changes only marginally with respect to the original model version. Frontal production and production in deeper stable stratified waters increased slightly, while production near the coast was slightly decreased. The model is thereby capable of resolving tidal influences on primary production via potentially competing processes. Tidal mixing releases nutrient limitation, thus fostering NPP, but tides also cause the resuspension and mixing of suspended matter into the euphotic zone, which reduces light availability in the water column, thus reducing NPP. In addition to relevant bottom-up processes, the model also resolved phytoplankton–zooplankton feedbacks and vertical oxygen and temperature profiles, which alter the remineralization of organic matter and consequently nutrient cycling.

Besides organic matter contribution to light shading, in the coastal area, inorganic SPM also has the potential to filter light and reduce primary production. We do not consider a dynamic coupled SPM modeling approach but consider a simplified consideration of inorganic SPM through implementing the background attenuation. To address the uncertainties related to SPM, we tested the effect of inorganic SPM on our findings with help of an additional numerical simulation, where we implemented a climatological SPM field (daily resolution, with 31 vertical layers in the original dataset) (Große et al., 2016; Heath et al., 2002) and added the SPM's contribution to the light attenuation scheme. Details of the inorganic SPM dataset and implementation are given in Appendix C. The results confirmed the validity of our assumption that the spatial variability of SPM can be neglected for the sensitivity study performed here. Despite the existing effect of inorganic SPM on light conditions and spatial variability of inorganic SPM, there is only minor sensitivity found for the case studies of tidal vs. non-tidal forcing, and Eq. (1) can be considered as a proper parameterization within the context of our study.

The ability to properly resolve intensified frontal production and the consideration of key processes influencing light and nutrient limitation related to tidal forcing make ECOSMO an appropriate tool to assess tidal impacts on NPP in the spatially highly diverse North Sea. As already stated in Daewel and Schrum (2013), ECOSMO estimates of annual NPP in the North Sea (Fig. 2) are at the lower edge of what has been simulated for the area (Holt et al., 2012; van Leeuwen et al., 2013). The relatively low estimates mainly appear in the northern North Sea (NNS), where primary production is estimated to be approximately $125\,\mathrm{gC\,m^{-2}\,year^{-1}}$ based on observations (Van Beusekom and Diel-Christiansen, 1994), and on the European continental coast, where NPP observations range between 199 and $261\,\mathrm{gC\,m^{-2}\,year^{-1}}$ (Joint and Pomroy, 1993). The simulation fits well with observation-based estimates of NPP on the British coast of approximately $75$–$79\,\mathrm{gC\,m^{-2}\,year^{-1}}$ (Joint and Pomroy, 1992) and primary production estimates of 100 and $119$–$147\,\mathrm{gC\,m^{-2}\,year^{-1}}$ in the central parts of the North Sea and at Dogger Bank, respectively (Joint and Pomroy, 1993).

## 2.2 Model setup

A detailed description of the model setup was given by Daewel and Schrum (2013); therefore, we will only provide a brief overview of the forcing data used for the model simulation, particularly emphasizing the changes made to the previously described setup. These changes mainly concern the river discharge and nutrient load data sources. The simulation was initialized in 1948 using climatological data from the World Ocean Atlas (WOA) (Conkright et al., 2002) for nutrients and observational climatology for temperature and salinity (Janssen et al., 1999). The full simulation period en-

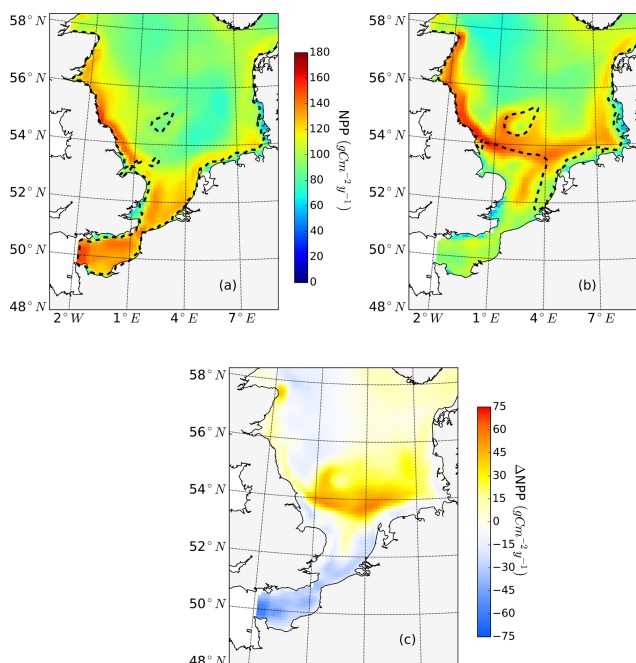

**Figure 2.** Mean annual net primary production for the analyzed period (1990–2015) of the non-tidal scenario **(a)**, tidal scenario **(b)** and the difference in the mean annual NPP of both scenarios **(c)**. Dashed lines indicate the boundary between stratified (off-shore) and unstratified (near-shore, Dogger Bank) regions. The criterion for stratification is that squared buoyancy frequency $N^2$ remains higher than 0.013 ($\mathrm{s^{-2}}$) for more than 60 d per year on average.

compasses 68 years, ending in 2015, and is forced with atmospheric boundary conditions provided by the NCEP/NCAR reanalysis (Kalnay et al., 1996). Additional forcing data include wet deposition for nitrogen, which were prescribed using data from a Community Multiscale Air Quality (CMAQ) model (Matthias et al., 2008), and boundary values for nutrients, temperature and salinity at the open boundaries to the North Atlantic, for which we used the same climatological data as those used for the initial conditions. For salinity, additional annual anomalies were retrieved from observational data available at the ICES (International Council for the Exploration of the Sea) database (http://www.ices.dk, last access: 30 November 2016). An updated set of river runoff and nutrient load data was applied with more complete river forcing data coverage for the North Sea and Baltic Sea. A multitude of data were provided by Sonja van Leeuwen (Royal Netherlands Institute for Sea Research, personal communication, 2016) containing the following datasets: UK data were processed from raw data from the Environment Agency (England and Wales, contains Natural Resources Wales information ©Natural Resources Wales and database rights), the Scottish Environment Protection Agency (Scotland), the Rivers Agency (Northern Ireland) and the National River Flow Archive. French water quality data were provided by

Agence de l'eau Loire-Bretagne, Agence de l'eau Seine-Normandie, OSUR web Loire-Bretagne and SIEAG (Systeme d'information sur l'eau du bassin Adour Garonne), while daily flow data were obtained from Le Banque Hydro (http://www.hydro.eaufrance.fr/, last access: 23 April 2019). German and Dutch riverine data were provided by the University of Hamburg (Pätsch and Lenhart, 2004). Norwegian water quality data were provided by the Norwegian Water Resources and Energy Directorate (NVE), with daily flow data supplied by the Norwegian Institute for Water Research (NIVA). Danish water quality data were provided by the National Environmental Research Institute (NERI). Water quality data for Baltic rivers were provided by the University of Stockholm and the Baltic Nest. Furthermore, nutrient status and freshwater runoff information in the southern and eastern Baltic Sea was supplemented by data from the Balt-HYPE model (Arheimer et al., 2012; Lindström et al., 2010). Nutrient loads from Danish waters were provided by Marie Maar (personal communication, 2016) and were similar to the forcing data used for the HBM-ERGOM simulation (Maar et al., 2016). These data stem from a national monitoring program (Windolf et al., 2011, 2012) and from the hydrological Denmark model, which provides runoff calculations for ungauged areas of Denmark (Henriksen et al., 2003).

We selected a relatively short time period (1990–2015) for our analysis to assure a long enough spin-up time that accounts for the characteristic long timescales of the North Sea–Baltic Sea system (Daewel and Schrum, 2013). The period from 1990 to 2015 will hereafter be called the analyzed period. Tidal cycles with long periods, such as the nodal and elliptical cycles, although considered in the forcing via nodal corrections of partial tide amplitudes and phases (see Sect. 2.3), are not targeted in this study.

## 2.3 Tidal forcing and scenarios

Sea surface elevation was prescribed at the open boundaries, with a time step of 20 min. Daily mean sea surface elevation data were taken from a diagnostic model simulation for the wider northeast European Shelf (Backhaus and Hainbucher, 1987) and also forced with the NCEP/NCAR reanalysis. In addition, tidal elevations were calculated from tidal constituents provided by the German Federal Maritime and Hydrographic Agency (Federal Maritime and Hydrographic Agency, Deutsches Hydrographisches Institut, 1967). Nodal corrections were implemented in the calculation of tides to represent the long-term variation in lunar nodes. For the standard tidal scenario, partial $M_2$ tide (principle lunar tide) and $S_2$ tide (principle solar tide) (Thomson and Emery, 2014) were considered; we hereafter call this scenario the tidal scenario. To evaluate the contribution of the spring–neap tidal cycle, a tidal scenario using only the $M_2$ partial tide, called the $M_2$ scenario, was simulated and discussed in comparison to the tidal scenario. To quantify the overall impact of tidal forcing, a scenario without tidal forcing at the open boundary was simulated to yield the non-tidal reference state of the system (non-tidal scenario).

## 2.4 Postprocessing of model results

The responses of ecosystem productivity to tidal forcing were assessed by comparing the annual mean NPP during the analyzed period between the tidal and non-tidal scenarios (tidal scenario minus non-tidal scenario). Furthermore, we disentangled processes that might contribute to variations in NPP, such as the seasonality of spatial patterns in limitation factors (nutrients vs. light), spring bloom phenology, the impacts of the spring–neap cycle on NPP variability and the contribution of subsurface production to the overall NPP. We quantified these processes using subdomains and further made comparisons between scenarios, emphasizing spatial variability and the seasonal cycle.

### 2.4.1 Subdomain division and identification of representative grid cells for process based analysis

The pre-division of the area into subdomains is based on a combination of geographic location, bathymetry and the local responses of NPP to tidal forcing (increase, decrease). First, SNS and NNS were divided by the 65 m isobath. In the SNS, areas with positive and negative NPP response to tides were separated (Fig. 2). The negatively responding area in the SNS was further geographically divided into the English Channel (EC, south of 52° N) and an area along the continental coast (neg. SNS). In the NNS, the area of the Norwegian Trench (NT) characterized by a water depth deeper than 200 m was separated. The remaining region of the NNS was further divided based on the response of NPP to tidal forcing. The area along the eastern British coast (BC), which shows elevated NPP in response to tides, was separated from the negative responding area in the middle of NNS (deep NNS) (Fig. 2). In the east of the NNS, a separate area with mild increase of NPP was identified (low-sen. NNS). Based on this pre-division of subdomains, we identified the most representative grid cell within each subdomain using correlation analysis (Eliasen et al., 2017) (Fig. A1 in Appendix A). To identify the most representative grid cell location in each subdomain, we first produced a time series of the NPP differences between the non-tidal scenario and tidal scenario for each grid cell. Subsequently, we estimated, for each of the grid cells, the correlation to the time series of the other grid cells within the same pre-divided subdomain. The grid cell with the highest correlation coefficient to all other grid cells in each subdomain was selected as the most representative point for further analysis.

### 2.4.2 Quantification of key processes controlling the spring bloom

The peak amplitude and the onset time of the spring bloom for the different scenarios were compared. The onset of the spring bloom is defined here as the day when the daily vertically integrated NPP reaches its maximum prior to the spring maximum in diatom biomass (Fig. A2) (Sharples et al., 2006). Diatom time series were preprocessed by a 15 d running mean to remove short-term maxima induced by the spring–neap tidal cycle (Sharples et al., 2006). To further disentangle mechanisms resulting in spring bloom phenology differences among the scenarios, we quantified potentially related biological and physical factors relevant for spring bloom dynamics, such as the zooplankton biomass prior to the onset of the spring bloom, light conditions and development of stratification, for each grid cell.

In particular, (i) the vertically averaged zooplankton biomass in the winter season (January and February) was considered a proxy for potential grazing pressure at the beginning of the growth season. (ii) The integrated value of the light-limiting term in the upper 50 m of the water column was used to estimate the light conditions for phytoplankton growth. To quantify the time when the light was sufficient for phytoplankton growth in each year, we estimated the date when the integrated light-limiting term exceeded 0.85 for 3 consecutive days. (iii) The stratification was recognized as the critical temperature difference ($\Delta T$) between surface layers and layers below exceeding 0.5°. Similar methods have been used in many other studies (Gong et al., 2014; Karl and Lukas, 1996; Richardson et al., 2002). For the identification of the onset time of stratification, $\Delta T$ exceeding 0.5 °C for 3 consecutive days is required. The time window (3 d) (Sharples et al., 2006) was chosen to filter out short-lived stratification variations and the day–night heating/cooling cycle. The mixed layer depth is defined as the thickness of the surface mixed layer, ranging from surface to pycnocline. (iv) The averaged mixed layer depth in May was used as a measure for stratification depth.

The onset of the spring bloom, the first day of the year with stratification and the first day of the year with sufficient light conditions were identified for each grid point for every simulated year; subsequently, the percentage of years in which those time identifiers were advanced or delayed in the tidal scenario compared to that in the non-tidal scenario as a response to tidal forcing was estimated for every grid cell. The tidal induced increase/decrease of winter zooplankton biomass and of peak spring bloom amplitude were also estimated for each grid cell and each year. Using those indexes, we obtained the spatial pattern for the percentage of years with (1) higher spring bloom amplitude, (2) later onset of the spring bloom, (3) later onset of stratification, (4) deeper mixed layer depth, (5) later occurrence of sufficient light conditions for building phytoplankton biomass and (6) higher

concentration of winter zooplankton biomass in response to tidal forcing (tidal scenario vs. non-tidal scenario).

Furthermore, we studied the changes in the spring bloom phenology in response to the spring–neap tidal cycle (i.e., whether spring or neap tide promote/hinder NPP). Considering that several spring–neap cycles may take place during the spring bloom development, we studied the NPP difference during of spring bloom development between the tidal scenario and $M_2$ scenario in relation to the spring–neap tidal phase. The period of spring bloom development was defined as the time period with an increase in NPP from 12.5 % to 87.5 % of the maximum NPP. During this time period, we identified the occurrences (within a time window of one fortnight cycle) of positive/negative maxima of the NPP difference and temporally related the day of maximum difference to the adjacent day of the spring tide. This enabled us to evaluate the impact of spring–neap tidal cycles on spring bloom phenology.

### 2.4.3 Quantification of limiting pattern of phytoplankton growth: light vs. nutrients

In ECOSMO, NPP is estimated as the sum of net primary production for all phytoplankton functional groups (Eq. 2, denoted by $j$). For each functional group, the NPP is calculated by multiplying the maximum growth rate specified for the functional group ($\sigma_j$) by the minimum value ($\varphi_j$) of all limiting terms ($\theta$) (Liebig's law, de Baar, 1994) and the prevailing amount of phytoplankton biomass (standing stock, $C_j$) (Eq. 2). The limiting term ($\theta$) for each growth resource is derived from the Monod equation (Monod, 1942), using the concentration of each growth resource ($\beta$) (Si: silicate-only for diatom growth, N: nitrogen, P: phosphorus, L: light) and the specific half-saturation constant ($h$) (Eq. 3). Further details of the nutrient-limiting terms are given in Daewel and Schrum (2013). We hereafter call the minimum value of all limiting terms $\varphi$ (Eq. 4) the limiting value. The limiting value quantifies the availability of growth resources with a range of 0–1. The closer the value is to 1, the more sufficient the resource is. Additionally, we identified the most limiting factors for each phytoplankton type ($\varphi_j$) (N, P and L for flagellates; Si, N, P and L for diatoms).

$$\text{NPP} = \sum_{j=1}^{3} \sigma_j \varphi_j C_j \tag{2}$$

$$\theta = \beta/(\beta + h) \tag{3}$$

$$\varphi = \min\left(\theta_{\text{light}}, \theta_{\text{N}}, \theta_{\text{P}}, \theta_{\text{Si}}\right) \tag{4}$$

We analyzed the limiting value to represent the environmental conditions of phytoplankton growth and the spatial and temporal dynamics of the most limiting factor.

### 2.4.4 Vertical distribution of phytoplankton: detection of subsurface maximum layer

The mixing intensity in the water column controls the distribution of phytoplankton and nutrients. As suggested by previous studies, phytoplankton may develop high subsurface concentrations in layers of low turbulence such as the pycnocline; production continues locally in low-turbulent zones as long as the growth requirements of nutrients and light are balanced (Cullen, 2015). In the stratified season, we differentiated the NPP generated in the surface layer (above 15 m) from that in the subsurface layers, as a subsurface biomass maximum (SBM) emerged. The SBM was defined by its width, which was small compared to the water depth, and was persistent in both time and space (Dekshenieks et al., 2001). In this study, we regarded layers deeper than 15 m as the subsurface. As an SBM necessarily includes local peaks, we first selected the depth at which the first-order derivative of biomass changed from positive to negative in the vertical biomass profile as a potential location for an SBM peak. To further identify the boundaries of the potential SBM, different strategies were applied depending on the number of vertical layers on either side of the potential SBM peak. If there were more than five vertical layers on either side of the potential SBM peak, the vertical layer with the local maximum in the second-order derivative on each side of the potential SBM peak was recognized as the boundary of the SBM layer (Benoit-Bird et al., 2009). Otherwise, the adjacent layers were assumed to confine the potential SBM. The SBM peak could be no shallower than 20 m. We estimated the local background biomass value by linearly interpolating the biomass values of the upper and lower edges to the depth where the peak in biomass emerged. If the peak maximum biomass exceeded a value 1.5 times higher than the estimated background biomass in the respective water column, the local vertical plankton biomass maximum was considered an SBM. Similar methods which have been applied to analyze phytoplankton (chlorophyll $a$) vertical profiles in the German Bight and more details were laid out in Zhao et al. (2019).

### 2.4.5 Identification of representative grid cells for spring–neap cycle impacts

In addition to tidal forcing, atmospheric forcing and bathymetry modulate stratification (e.g., Van Leeuwen et al., 2015) and production pattern (Daewel and Schrum, 2017). Consequently, tidal impacts on stratification and primary production are subject to spatial–temporal variability. Furthermore, non-linear interactions among tidal constituents are pronounced in shallower waters, as suggested by Backhaus (1985) in inshore areas for the German Bight and Danish coast. Although we preliminarily estimated the influence of the spring–neap tidal cycle via the difference in NPP between the tidal scenario and the $M_2$ scenario, related responses would not necessarily be visible in a fortnightly cycle. To better associate the variation in NPP with the spring–neap tidal cycle, we identified specific grid cells where both currents and biochemical factors displayed a distinguishable spring–neap cycle. Those locations were identified by using the estimated squared coherence between the power spectra (SCPS) of currents and NPP (Stoica et al., 2005; Welch, 1967). By adopting the SCPS method, we were able to select representative grid cells where both NPP and velocity showed obvious spring–neap cycles.

## 3 Results and discussion

### 3.1 Spatial changes in mean production

The average annual NPP and the difference in NPP between the tidal and non-tidal scenarios are shown in Fig. 2. The area-averaged NPP increases slightly from 100.7 to 103.2 g C m$^{-2}$ year$^{-1}$ when tidal forcing is applied (Table 1); however, high spatial diversity in the sensitivity to tidal forcing is shown. Generally, estimated tidal impacts on NPP are highest in the stratified shallow North Sea, with a maximum response of up to 60 g C m$^{-2}$ year$^{-1}$ (Fig. 2c). In the non-tidal scenario, high productivity is restricted to the near-shore shallow regions along the British coast and the European continental coast (Fig. 2a), which are the main regions where the euphotic zone reaches the bottom and nutrient remineralization fosters production throughout the year. The primary production at the coast is additionally supported by estuarine-type baroclinic circulation in summer, which transports detritus and nutrient-rich bottom water towards the coast (Ebenhöh et al., 2004; Geyer and MacCready, 2014; Hofmeister et al., 2017). Tides cause a significant reduction in stratification in the shallow near-coastal areas of the North Sea and in the EC at Dogger Bank and south of Dogger Bank and foster the development of tidal mixing fronts. Consequently, the production pattern changes notably when tidal forcing is considered. The primary production maximum is shifted further offshore towards the frontal region (Fig. 2b). Large areas of the SNS, including Dogger Bank, eastern BC and the Danish coast in the east, together with the NT, exhibit an increase in NPP when tidal forcing is prescribed. The shallow near-coastal areas in the south and the deeper areas in the NNS show a negative response of NPP to tidal forcing. A stronger negative response is observed in the highly dynamic EC (Fig. 2c). The NPP of Dogger Bank and the tidal mixing front area south and southeast of Dogger Bank responds the strongest to tidal forcing, with a mean change in NPP of up to 60 g C m$^{-2}$ year$^{-1}$, nearly doubling local production. The amplitudes of the decreases in NPP in the negatively responding area are smaller than those of the increases in NPP, with amplitudes no more than 40 g C m$^{-2}$ year$^{-1}$ (Fig. 2c); the largest amplitudes are in the EC. The intensity of this difference might be slightly sensitive to the consideration of inorganic SPM (see Appendix C).

**Table 1.** Average annual NPP and relative difference between the tidal and non-tidal scenarios in each subdomain and in the entire North Sea.

| Subdomain | Non-tidal scen. NPP (gC m$^2$ year$^{-1}$) | Tidal scen. NPP (gC m$^2$ year$^{-1}$) | Rel. diff (%) |
|---|---|---|---|
| EC | 125.2 | 97.2 | −29 % |
| neg. SNS | 114.5 | 101.6 | −13 % |
| pos. SNS | 93.3 | 118.8 | 21 % |
| BC | 121.0 | 135.3 | 11 % |
| deep NNS | 93.8 | 82.6 | −14 % |
| NT | 97.7 | 106.4 | 9 % |
| non-sen. NNS | 92.3 | 94.5 | 2 % |
| Total | 100.7 | 103.2 | 3 % |

The tidally induced change in NPP is associated with variations in the spatial distribution of the main limiting resources (limiting pattern) (Fig. 3). Generally, in the tidal scenario, the area experiencing nutrient limitation decreases due to the enhanced mixing of inorganic nutrients into the euphotic zone, especially in the shallow North Sea where the bottom and surface mixed layer interact with each other. Simultaneously, light limitation increases. The predominantly light-limited regions, which are restricted to the shallow coastal regions in the non-tidal scenario (Fig. 3a), expand offshore in the tidal scenario (Fig. 3b). Tidally induced resuspension and mixing of particulates and DOM into the euphotic zone result in dominant light limitation in almost the entire shallow North Sea (below 50 m depth) (Fig. 3b). In contrast, in the surface layers of the stratified area, summer nutrient limitation is predominant, and the limiting value remains below 0.3 in both scenarios. The change from nutrient to light limitation in the SNS changes the limiting value to > 0.4 in the tidal scenario, allowing better resource exploitation in these areas and sustaining NPP during summer.

The subdomain-division method described in Sect. 2.4.1 identifies seven different subdomains (Fig. 4) that show characteristic responses to tidal forcing. Based on the division and the point-wise correlation of NPP variations in each subdomain (Fig. A1), representative grid cells were selected to study the mechanisms underlying the spatial variability of tidal responses in detail. Areas with correlation coefficients higher than 0.3 occupied at least 53 % of each subdomain, comprising 77 % of the entire study area. This indicates that the division effectively explains the spatial diversity of the system with respect to the tidally induced changes in NPP and the predominantly inherent similarity within each subdomain. The seven identified subdomains are listed below (Fig. 4):

1. *The English Channel (EC; dark blue).* This area is characterized by an early onset of the spring bloom, strong mixing due to tidal stirring and shallow bathymetry. The EC is the most productive area in the non-tidal scenario (Fig. 2a), with a mean NPP above 120 gC m$^{-2}$ year$^{-1}$.

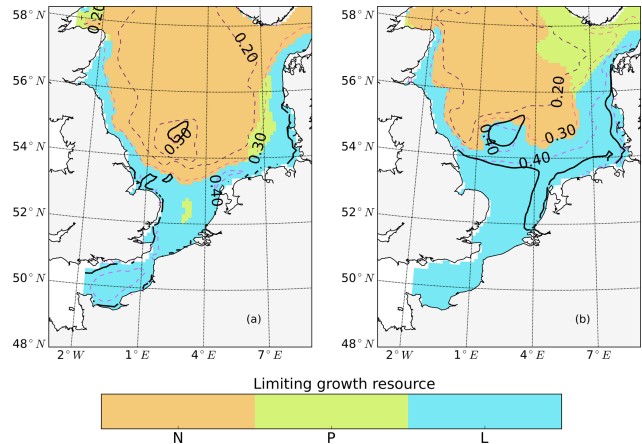

**Figure 3.** Mean values of the most limiting resources (N: nitrogen, P: phosphorus, L: light) in the surface layer for July (averaged for the analyzed period; 1990–2015) for the non-tidal scenario **(a)** and tidal scenario **(b)**. The limiting value (derived from Liebig's law) is indicated by dashed contour lines. Stratified and unstratified areas are separated by black lines (for definition, see Fig. 2).

2. *Negatively responding southern North Sea (neg. SNS; blue).* The neg. SNS is separated from the EC by 52° N and from the positively responding area in the southern North Sea. The neg. SNS characterizes the permanently mixed area in the shallow water near the coast.

3. *Positively responding southern North Sea (pos. SNS; light blue).* This area includes the frontal regions that were identified as the areas with the highest responses in NPP (Fig. 2).

4. *Eastern British coast (BC; green).* This area is a highly productive, positively responding inshore region of the eastern British coast.

5. *Deeper northern North Sea (deep NNS; yellow).* The deep NNS region coincides with areas of seasonal stratification and the lowest annual NPP in the tidal scenario

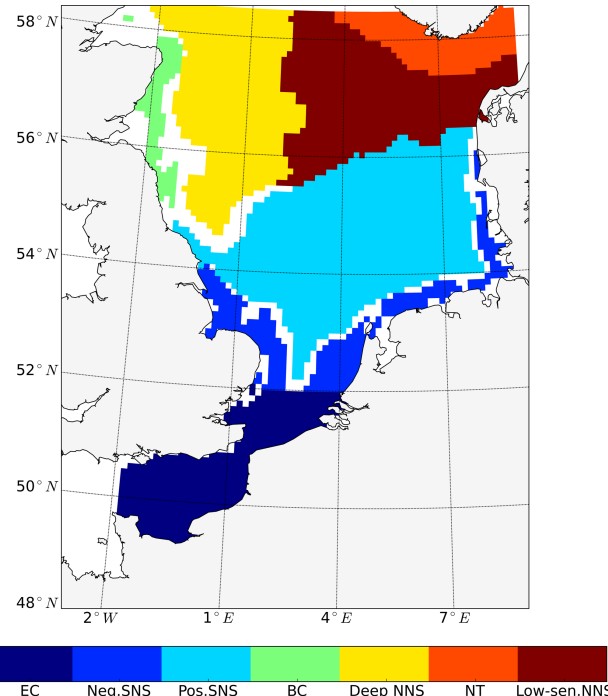

**Figure 4.** Process-oriented subdomain division of the North Sea based on tidally induced changes in net primary production and bathymetric characteristics (EC: English Channel; neg. SNS: negatively responding southern North Sea; pos. SNS: positively responding southern North Sea; BC: eastern British coast; deep NNS: deeper northern North Sea; NT: Norwegian Trench; low-sen. NNS: low-sensitivity northern North Sea). Areas with an absolute variation in NPP less than $5\,\mathrm{gC\,m^{-2}\,year^{-1}}$ are excluded, except for the low-sen. NNS areas.

(Fig. 2). In this area, a slight decrease in NPP is estimated when tidal forcing is considered.

6. *The Norwegian Trench (NT; orange).* This represents the area off the Norwegian coast which is strongly impacted by the low saline outflow from the Baltic Sea. The NT shows a slight increase in NPP due to tidal forcing (Fig. 2).

7. *Low-sensitivity area in the northern North Sea (low-sen. NNS).* The magnitude of the response of NPP to tidal forcing here is below $5\,\mathrm{gC\,m^{-2}\,year^{-1}}$. This subdomain is influenced by two amphidromic points in the eastern North Sea, with tidal amplitudes of the $M_2$ partial tide generally below 0.5 m.

Some narrow transient zones between the positively responding areas and negatively responding areas are shown in white in Fig. 4. These transient zones with an absolute variation in NPP less than $5\,\mathrm{gC\,m^{-2}\,year^{-1}}$ are excluded from the following analyses. Changes in NPP in response to tidal forcing for each subdomain are listed in Table 1.

The subdomain division corresponds well with the regional characteristics of $M_2$ tidal energy dissipation rates, as suggested by the simulation study of Davies et al. (1985). The EC subdomain includes the areas with the highest tidal energy dissipation rates, which exceed $1000\,\mathrm{J\,cm^{-2}\,s^{-1}}$ (Davies et al., 1985). In most of the neg. SNS and some parts of the EC, the tidal energy dissipation rates are in the range of $100$–$1000\,\mathrm{J\,cm^{-2}\,s^{-1}}$. In the pos. SNS, the BC and part of the deep NS, tidal energy dissipation rates range from 10 to $100\,\mathrm{J\,cm^{-2}\,s^{-1}}$. The low-sen. NNS and NT are located in the area with tidal energy dissipation rates below $10\,\mathrm{J\,cm^{-2}\,s^{-1}}$. The strong tidal energy in the SNS destabilizes stratification, as also revealed by the subdivision based on stratification patterns presented by Van Leeuwen et al. (2015). Our neg. SNS and EC subdomains coincide with permanently mixed regions defined in the above study; in addition, the defined BC correlates with mixed or temporally stratified belts along the eastern British coast, as suggested by Van Leeuwen et al. (2015). The subdomains identified in the NNS coincide with seasonally stratified areas in the aforementioned study. However, the majority of pos. SNS, which shows the strongest response to tidal forcing, could not be identified with the method of Van Leeuwen et al. (2015) due to the variable stratification in these frontal areas induced by the spring–neap cycle, wind forcing, river runoff and air temperature (Dippner, 1993; Schrum et al., 2003a; Sharples and Simpson, 1993). The subdomains also agree well with subdomains previously identified by Otto et al. (1983) TS1. Compared to the ICES subdivisions, which were determined considering biochemical and hydrographical characteristics (Otto et al., 1983), the four northern subdomains in our study coincide with regions where the gross water mass influx is mainly influenced by Atlantic water inflow. In contrast, for the three subdomains in the south, the influence of wind is more important for water mass exchange (Siegismund, 2001).

## 3.2 Characteristic seasonal changes

Out of the seven subdomains (Fig. 4), we selected three representative subdomains for further analysis of the changes in seasonality of NPP and the respective associated mechanisms. The neg. SNS represents the area along European continental coast where strong tidal forcing leads to permanent mixing and the NPP decreases as a consequence of tidal forcing. The pos. SNS embodies the transient zone between the mixed and stratified water column and is characterized by the most significant positive response of NPP to tidal forcing. The deep NNS is characterized by stable seasonal stratification. Here, the bottom mixed layer and surface mixed layer are well separated; thus, tides have a limited impact on the euphotic zone. The averaged time series (1990–2015) for each subdomain and the time series of the vertical profiles of each most representative grid cell (see Sect. 2.4.1) are given in Figs. 5 and 6, respectively.

Please note the remarks at the end of the manuscript.

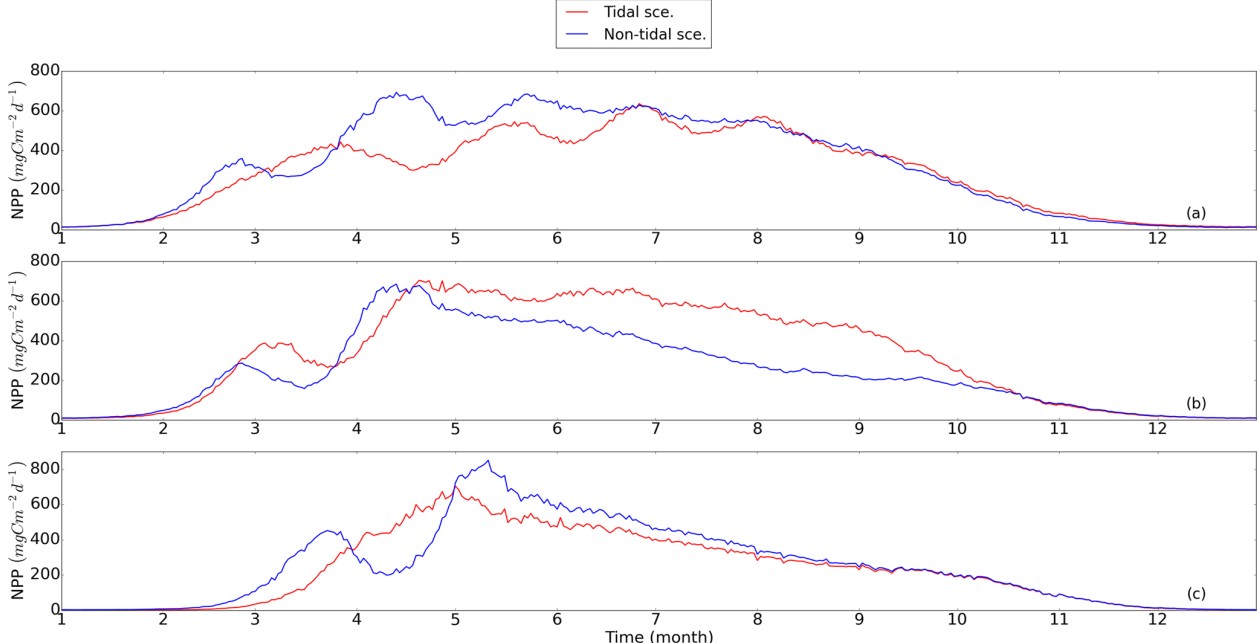

**Figure 5.** Time series of averaged NPP (blue: non-tidal scenario, red: tidal scenario) in subdomains; **(a)** neg. SNS: negatively responding southern North Sea, **(b)** pos. SNS: positively responding southern North Sea and **(c)** deep NNS: deeper northern North Sea. The NPP is averaged for the analyzed period (1990–2015).

In the neg. SNS (Fig. 5a), the spring bloom is delayed and strong fluctuations appear during the productive season in both scenarios (Figs. 5a and 6a, b, c, d). The pulses in NPP are probably due to predator–prey interactions and possibly modulated by advection. These pulses in NPP have previously been described by Tett and Walne (1995). The length of these fluctuations is slightly longer in the tidal scenario than in the non-tidal scenario, and changes in bloom initiation and the length of the quasi-periodic fluctuations generate positive–negative fluctuations in the NPP difference between both scenarios. We found no nutrient limitation in the water column in either scenario (Fig. 6a, b) and no significant changes in the limiting values (Eq. 4) (note: the minimum limiting value stems from light limitation), except for the slightly higher values in deep water column under the tidal scenario (Fig. 6c, d). This exception is likely caused by the downward mixing of shade-producing organic materials (e.g., phytoplankton, DOM and detritus), which leads to improved light conditions in the upper layer and better penetration. However, this result does not explain the negative NPP response in the area. Lower NPP in the tidal scenario than in the non-tidal scenario, especially in spring and early summer, results in an overall negative response in NPP. A likely reason for the reduction in NPP in the neg. SNS subdomain could be the tidally induced dilution of phytoplankton biomass in the euphotic zone in the shallow areas. The increased mixing in the tidal scenario dilutes the phytoplankton concentration in the upper, highly productive water layer (see vertical profiles of biomass in Fig. A4a, b) and consequently reduces the time

during which phytoplankton cells are exposed to high surface irradiance. Considering the small difference in the growth resources between the two scenarios (Fig. 6c, d), we mainly attribute the variation in NPP to the vertical distribution of standing stocks.

The most dominant change in seasonality as a consequence of tidal forcing in the seasonally stratified subdomains (pos. SNS and deep NNS) is the delay of the spring bloom in the tidal scenario (Fig. 5b, c). However, in the pos. SNS, this delay is only a few days long; in the deep NNS, this delay encompasses 1 month. Accompanying the delay, the amplitude of the spring bloom in the tidal scenario, especially in the pos. SNS, exceeds that of the non-tidal scenario. The spring bloom in the NS typically consists of diatoms, while after silicate depletion, flagellates dominate the summer production (McQuatters-Gollop et al., 2007; Schrum et al., 2006). Comparing the seasonality of NPP variation (Fig. 5b, c) with the annual averaged NPP deviation between scenarios (Fig. 2c), we found that the variation in NPP in summer is basically in phase with the direction of the NPP's response to tidal forcing for both the deep NNS and pos. SNS. Especially in the pos. SNS (Fig. 5b), summer blooms are higher in the tidal scenario than in the non-tidal scenario, with a maximum difference in July and August, fostered by weaker stratification and regular nutrient injections into the surface mixed layer due to tidally induced turbulence (Fig. 6f, h). Surface summer production is sustained throughout the summer at values of approximately $50\,\mathrm{mgC\,m^{-3}\,d^{-1}}$ and more in the upper 15 m (Fig. 6f), and light remains the

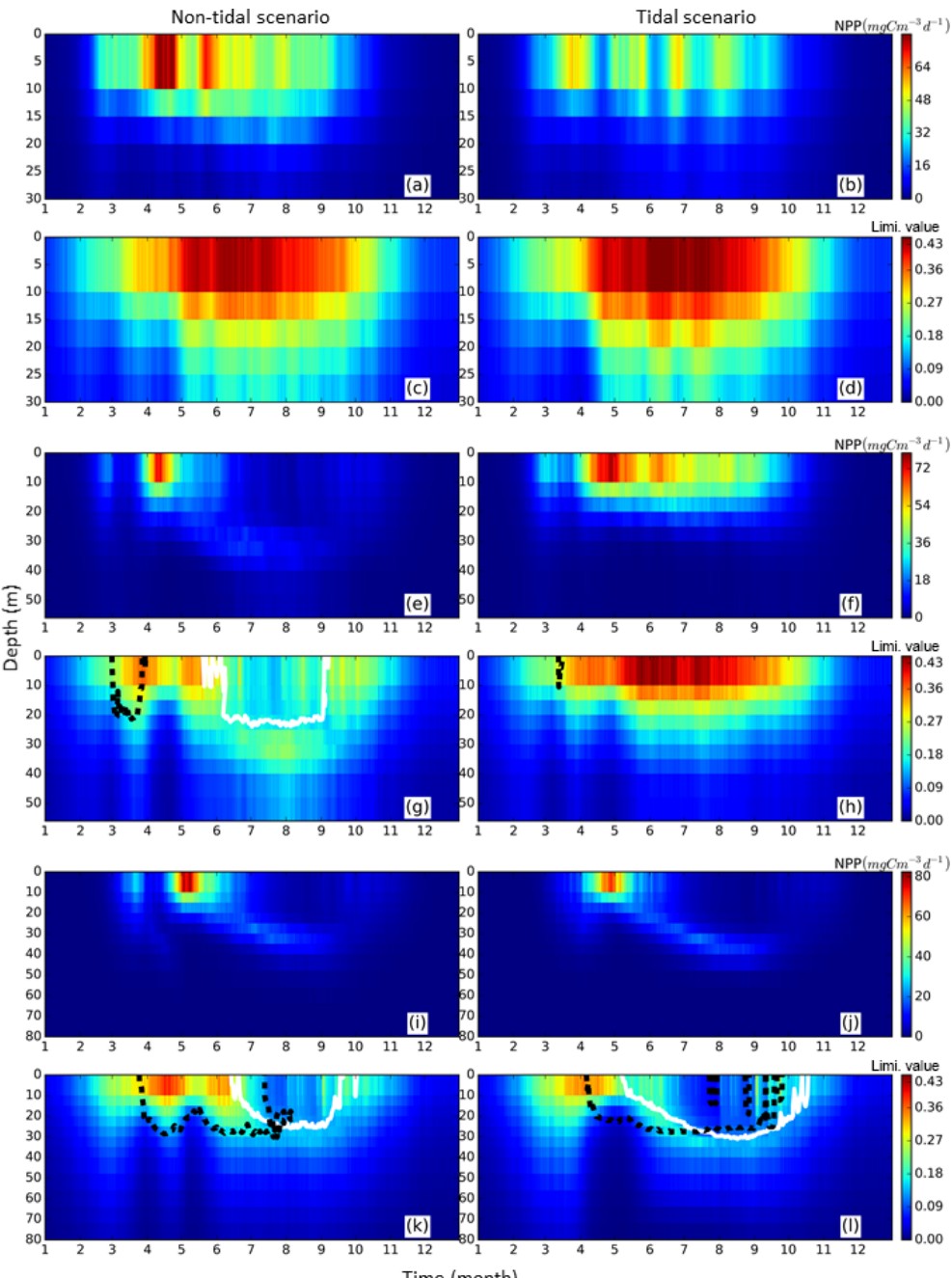

**Figure 6.** Time series of averaged (1990–2015) NPP vertical profiles (upper panels for each representative grid cell) and the limiting value (lower panels for each representative grid cell) for the tidal (right) and non-tidal (left) scenarios. NPP and limiting values are presented as the mean of each representative point for three subdomains, i.e., neg. SNS, the negatively responding southern North Sea (a–d), pos. SNS, the positively responding southern North Sea (e–h) and deep NNS, the deeper northern North Sea (i–l). Additionally, the depth above which a specific nutrient (silicate: solid black line, nitrogen: dashed white line) is limiting to NPP is given.

dominant limiting factor in the surface layer, except for a temporal silicate limitation after the spring bloom (Fig. 6h). In contrast, without tidal stirring, surface waters become nutrient depleted soon after the spring bloom in May. After silicate limitation, nitrogen limitation persists (Fig. 6g) in the

surface waters throughout the seasonal stratification, which results in the characteristic subsurface production in summer (Fig. 6e). Due to the weaker stratification and enhanced turbidity caused by tides, no SBM production occurs in this simulation. The nutrient supply advantage in the tidal sce-

nario persists until the beginning of October (Fig. 6f), when the water column in the non-tidal scenario is also mixed by atmospheric conditions, causing an increase in production at the surface. For the pos. SNS, the modulation of nutrient availability is the most important factor responsible for changes in NPP. The high biomass stays in the pycnocline during summer due to the weak mixing in the non-tidal scenario. In contrast, due to the weak stratification and strong mixing, generated high biomass is continuously mixed in the euphotic zone in the tidal scenario (Fig. A4c, d).

In the deep NNS, the influence of tides on NPP is relatively weak and mainly visible in summer (Fig. 5c). The deep NNS (Fig. 6i–l) is typically characterized by stable seasonal stratification and summer subsurface primary production in both the tidal and non-tidal scenarios. The delay of the spring bloom in the tidal scenario causes a quicker succession and consequently overlapping diatom and flagellate blooms (Figs. 6l, 5c). The productive period, which lasts nearly 3 months and includes two pulses of NPP in the non-tidal scenario (Fig. 6i), is shortened to 6 weeks in the tidal scenario (Fig. 6j). The NPP contributed from subsurface production is higher in the tidal scenario than that in the non-tidal scenario (Fig. 6i, j). Because of stratification and nutrient depletion, high biomass is confined to a region within the pycnocline in both scenarios. The SBM in the tidal scenario deepens because of mixed layer deepening due to tides (Fig. A4e, f).

In the other identified subdomains (results not shown), the changes in primary production basically follow the pattern explained above. In the EC subdomain, the tidal impact on production is comparable to that in the neg. SNS, whereas in the BC subdomain, nutrients are rarely the most limiting factors due to weak stratification, and the response can be compared to that in the pos. SNS. In the low-sen. NNS, where tidal dissipation is weak, the vertical distribution pattern of NPP in both scenarios is almost identical.

Our results indicate that, in principle, tidal stirring causes two major changes in the NPP pattern: (i) a change in the spring bloom phenology of some areas and (ii) an altered ratio between surface and subsurface production. Both features merit further discussion, which is given in the following paragraphs.

### 3.2.1 Changes in spring bloom phenology

As one of the most important biological events in the NPP annual cycle (Bagniewski et al., 2011; Sabine et al., 2004), the spring bloom requires specific attention. As shown by the time series analysis for some subdomains (Fig. 5) and the time series of profiles at the representative points (Fig. 6), the postponement of the spring bloom is a prevalent phenomenon when tidal forcing is applied. The changes in spring bloom phenology and the processes responsible for these changes, such as the delay in the onset of stratification, variations in light conditions, the mixed layer depth and winter

zooplankton concentrations (Fig. 7), were analyzed using the method outlined in Sect. 2.4.2.

In line with the distribution of tidal energy dissipation given by Davies et al. (1985), the spring bloom delay is robust in the SNS and along the British coast (Fig. 7a), while in the northeastern part of the North Sea, the spring bloom is delayed in no more than 50 % of all years. An increase in the peak spring bloom biomass (Fig. 7b) is mainly in areas with a positive response of NPP to tidal forcing (Fig. 2c) [CE1]. However, in some isolated locations in the negatively responding areas, such as the neg. SNS and EC, the spring bloom amplitudes are still higher in the tidal scenario than those in the non-tidal scenario in more than 50 % of the years. One potential reason for the spring bloom delay is a change in light conditions, especially in very shallow coastal, non-stratified areas where tidal stirring enhances resuspension in the water column (Fig. 7c). The onset of light conditions sufficient for phytoplankton growth in the well-mixed water column is delayed in the coastal areas of the southern and eastern boundary and in the shallower parts of Dogger Bank. However, the distribution of this impact does not explain the major patterns of changes in the spring bloom phenology. Tides also increase mixing and hence potentially prevent stratification in shallow water columns or delay the onset of stratification, as discussed previously by a number of authors (Bowden and Hamilton, 1975; Loder and Greenberg, 1986). Because tidally induced energy dissipation is cubically proportional to the strength of tidal currents (Simpson and Hunter, 1974), we can expect the strongest variation in stratification in regions with the strongest tidal currents, as observed along the British coast, in the EC and in the German Bight (Davies et al., 1985). This expectation is supported by earlier observations suggesting that the onset of the spring bloom is triggered by improved light conditions because of solar radiation and stratification (van der Woerd et al., 2011). The onset of stratification (Fig. 7e) in the tidal scenario is mainly delayed on the Scottish coast and the frontal areas of the SNS. Furthermore, the response of stratification to tidal forcing is more stable in the southwestern part (the Estuary of Humber, Dogger Bank) than in the southeastern part of the SNS (Fig. 7e). Apart from solar heating, the stratification in the southeastern part of SNS is additionally influenced by freshwater supplies from land and wind forcing (Jacobs, 2004; Ruddick et al., 1995; Schrum, 1997). Consequently, the variation in the onset of stratification is less clear in the southeastern part than in other parts of the SNS. In the NNS, the tidal wave propagation deepens the mixed layer depth (Fig. 7d), which similarly results in a later onset of the spring bloom, despite only weak changes in the onset of stratification. As a consequence of the thicker layer in which phytoplankton are mixed, the phytoplankton are less exposed to the favorable surface light conditions and will thus take longer to build up the spring bloom biomass.

Although the North Sea is in principle a bottom-up-controlled ecosystem, zooplankton predation is occasionally

Please note the remarks at the end of the manuscript.

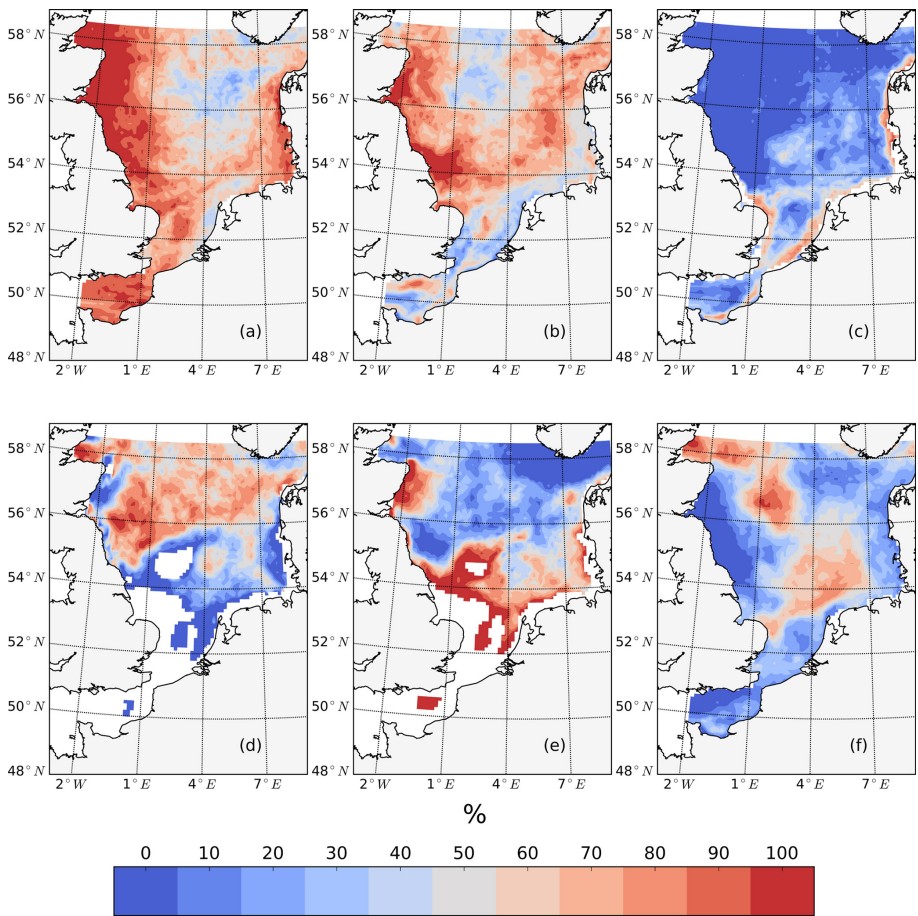

**Figure 7.** The percentage of years (1990–2015) in which specific processes potentially related to spring bloom phenology changed after considering tides. Changes include a later onset time of the spring bloom **(a)**, higher peak spring bloom biomass amplitudes **(b)**, a delay in the onset of light conditions in the water column sufficient for phytoplankton growth, as indicated by an integrated light-limiting term in the upper 50 m, exceeds 0.85 **(c)**, the deepening of the mixed layer depth in May **(d)**, a later onset time of stratification, which occurs when the maximum vertical temperature difference in the water column exceeds 0.5 °C for 3 consecutive days **(e)** and a higher concentration of overwintering zooplankton biomass **(f)**.

an important process controlling NPP (Daewel et al., 2014). In early spring, even under favorable growth conditions, the spring bloom will only initiate until production exceeds the loss due to grazing (George et al., 2015; Martin, 1965). This grazing pressure is basically correlated with the overwintering zooplankton stock. Based on our results, increases in the winter zooplankton biomass and delays in the spring bloom coincide only in the frontal region of the SNS and central NNS. Therefore, we conclude that the delay in spring bloom by tides is mostly due to bottom-up control.

The spatial pattern given in Fig. 7 shows that the delayed onset of the spring bloom in the tidal scenario may mainly be attributed to deteriorated light conditions in the shallow well-mixed area (Fig. 7c) and changes in the stratification of seasonally stratified areas, such as delays in the development of stratification (Fig. 7e) or the deepening of the upper mixed layer (Fig. 7d). Although the predator biomasses are higher prior to spring bloom in some areas, enhanced grazing

pressure at the beginning of the bloom period does not seem to be the main mechanism delaying the onset of the spring bloom (Fig. 7f), although we assume this pressure plays an additional role in the central NNS and frontal regions.

### 3.2.2 Changes in subsurface production in stratified season

To further quantify the magnitude of the changes in surface and subsurface production during the stratified season, we separated NPP vertically into upper-layer production (above 15 m) and production in the SBM layer and compared the results between scenarios (Fig. 8), using the mean annual value for the analyzed period (1990–2015). At the stratified side of the frontal zones (pos. SNS), the surface production response of NPP is positive almost everywhere, with a maximum reaching $+50 \, \text{gC m}^{-2} \, \text{year}^{-1}$ (Fig. 8b) at south of Dogger Bank. In contrast, the changes in response to tidal

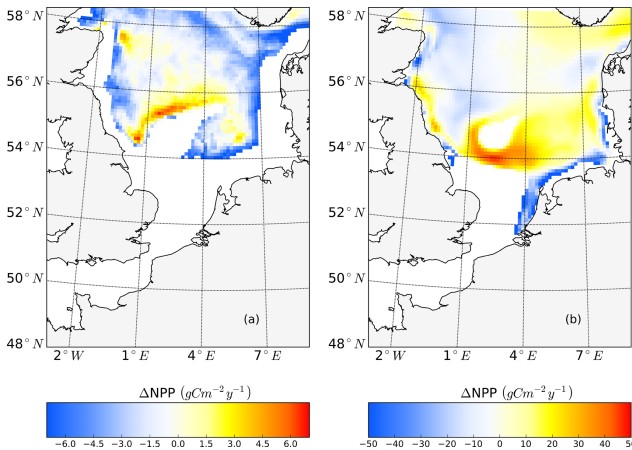

**Figure 8.** Mean difference in the NPP between the tidal and non-tidal scenarios generated within the SBM layer **(a)** and in the surface layer (above 15 m) **(b)**. The results are averaged for the stratified season. Areas with SBM mean occurrences of less than 10 d per year are excluded in panel **(a)**. Areas with stratification (squared buoyancy frequency $N^2 >= 0.013$ (s$^{-2}$) averaged less than 60 d per year are excluded in panel **(b)**.

forcing within the SBM show both negative and positive responses around Dogger Bank (Fig. 8a). A positive response to tidal forcing, which is generally 1 order of magnitude smaller than the increased amplitude of NPP in the surface layer, occurs only at the northern edge around Dogger Bank and the deeper part of the German Bight. A similar pattern with a strong positive response to tidal forcing at the surface and a negative response in the SBM appears in the BC area. In line with former studies in the North Sea, the NPP in the upper layer dominates the whole production budget (van Leeuwen et al., 2013). Although the expansion and duration times of the SBM decrease due to tidal forcing, e.g., in the inshore areas along the BC and at the Danish coast (Fig. A3b, d), tidal forcing promotes NPP within the SBM in some areas, especially at the northern edge of Dogger Bank. Observational studies suggested that the productive areas at the edge of Dogger Bank are fueled by baroclinic circulation related to the front and the spring–neap adjustment (Pedersen, 1994). When considering an SBM duration of 110 d (Fig. A3c) at the northern edge of Dogger Bank, the average daily NPP (deduced from the annual NPP; Fig. A3a) is approximately 239 mgC m$^{-2}$ d$^{-1}$, which corroborates the observation-based estimate of NPP (295 mgC m$^{-2}$ d$^{-1}$) calculated from measured oxygen surplus concentration data (Richardson et al., 2000).

In the NNS, the variation caused by tidal forcing in NPP is below 15 gC m$^{-2}$ year$^{-1}$ (Fig. 2c). In some parts of the deep NNS, the tidal forcing causes higher production in the SBM and lower production at the surface (Figs. 6i, j; 8a). Due to the decoupling between the surface and bottom mixed layers, the pycnocline acts as a barrier that keeps the stirred-up

nutrients below the pycnocline and sustains NPP in the SBM (Fig. 6i, k). Because the amplitude of NPP variations in the upper layers is 10 times higher than that in the SBM (Fig. 8), the overall response to tidal forcing is negative (Fig. 2c) in the deep NNS.

## 3.3   Impacts of the spring–neap cycle

The spring–neap tidal cycle introduces a fortnightly periodic change in tidal mixing, which has a significant influence along the British coast and in the English Channel (Fig. 9). The differences in current speed between the tidal and $M_2$ tidal scenarios vary over the spring–neap tidal cycle. The maximum spring–neap range of these differences is up to 0.3–0.6 m s$^{-1}$ (Fig. 9), indicating that a non-negligible change in turbulent kinetic energy is introduced to the water column via the spring–neap cycle. Here, we will provide model estimates on the spatial variability in the resulting response of the NPP to the spring–neap cycle and explore the potential mechanisms of these responses.

Annual NPP changes induced by the spring–neap cycle reach maximum values of up to 5 gC m$^{-2}$ year$^{-1}$ (Fig. 9). Although this amount is relatively small compared to the overall system productivity, the changes due to spring–neap dynamics could be very relevant locally and in specific time periods. An average positive response of NPP emerges in the southeastern part of the North Sea, in the English Channel and along the British coast (Fig. 9). The highest mean changes in NPP are found in the western part of Dogger Bank, in the English Channel and off the Scottish coast. In contrast, a negative response in annual production emerges off the Northumbrian coast and in the Southern Bight off the European continent (Fig. 9). The response of NPP to spring–neap tidal forcing is weak in early spring and winter (data not shown). Under mixed conditions or during periods of the establishment and decay of stratification, spring–neap tidal mixing can be overridden periodically by other mixing events (e.g., driven by wind); hence, pronounced irregularities in NPP responses to spring–neap tidal forcing are detected. A significant response of NPP to spring–neap tidal forcing is found for summer periods under stable stratification. To illustrate the basic mechanisms responsible for the response of NPP due to spring–neap tidal cycle, we present time series of the biomass, nitrate, NPP and turbidity (Eq. 1) profiles for two characteristic grid cells (selection described; see Sect. 2.4.5 TS2) that respond differently to spring–neap tidal forcing. The near-shore grid cell off the Estuary of Humber (EH, Fig. 9) shows a negative response, and a grid cell located at the frontal zone at the western edge of Dogger Bank (WDB, Fig. 9) responds positively to spring–neap tidal forcing. The model results are presented for a couple of selected successive spring–neap tidal cycles simulated for the year 2001 (Fig. 10).

The EH site, which is located further inshore compared to the WDB, is characterized by high turbidity. The increased

Please note the remarks at the end of the manuscript.

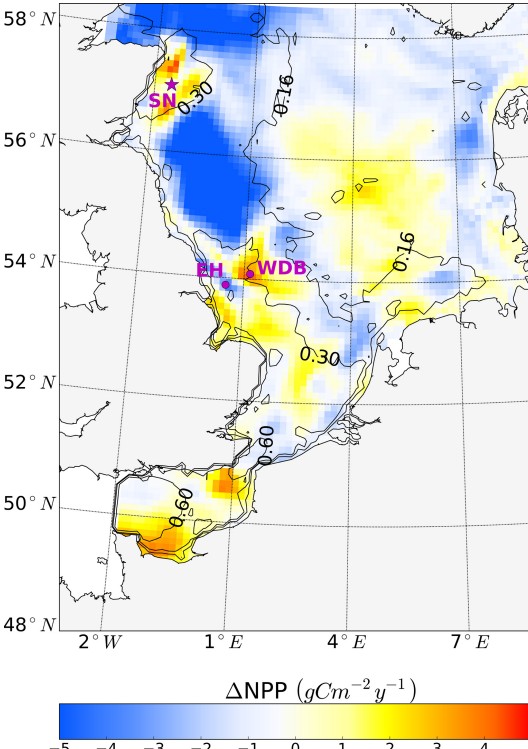

$\Delta$NPP $(gCm^{-2}y^{-1})$

**Figure 9.** Simulated annual mean NPP difference between the tidal $(M_2 + S_2)$ and $M_2$ scenarios, averaged for 1990–2015. Positive values depict higher NPP values in the tidal scenario $(M_2+S_2)$ than in the $M_2$ scenario. Contour lines indicate the estimated mean spring–neap cycle range of the tidal current speed difference between the tidal $(M_2 + S_2)$ and $M_2$ scenarios. The two magenta dots indicate the locations of two characteristic grid cells. One grid cell is close to the Estuary of Humber (EH), and the other grid cell is located more offshore at the western edge of Dogger Bank (WDB) (see Fig. 10). The magenta star shows the location of the grid cell used for the analysis of the advancement and delay of the spring bloom due to spring–neap tidal forcing (see Fig. 11).

nitrogen in the upper layers is in phase with elevated turbidity but in anti-phase with biomass and NPP. This phenomenon indicates that during spring tide, the process of phytoplankton biomass dilution (Fig. 10c) and shading due to the upward mixing of organic material (Fig. 10b) slows NPP in the upper mixed layer, resulting in a negative NPP response during spring phases (Fig. 10d). The elevated NPP reaches a maximum at the end of the neap phase (Fig. 10d), possibly because of the reduced vertical mixing. The decreasing turbidity in the neap phases, despite increases in phytoplankton biomass, reveals that suspended and resuspended organic material have a reduced impact on the surface light conditions during neap phases compared to the spring phases (Fig. 10b). In neap tidal phase, given less vertical mixing, phytoplankton cells remain in the lighted surface layer for longer time and access better light conditions; hence, the

available nutrients can be utilized for phytoplankton growth (Fig. 10c).

In contrast, the WDB site is typically characterized by seasonal stratification and summer nutrient (i.e., nitrate) depletion in the surface layer. However, as the WDB site is located in the frontal zone, relevant factors in this zone do not necessarily show the spring–neap fluctuation as clearly as those at the EH site. During spring tide, enhanced vertical mixing dilutes the phytoplankton biomass in the upper layer and redistributes biomass more evenly in the whole water column, resulting in less phytoplankton biomass in the upper layer (blue) and more biomass in the lower layer (red) compared to that in the $M_2$ scenario (Fig. 10g). Spring tidal forcing results in the replenishment of nutrients in the euphotic zone and a pulse of increased NPP follows spring tide mixing (Fig. 10e, h). The downward mixing of biomass into lower layers has no substantial negative effect on NPP during spring tide (Fig. 10g). As a consequence of nutrient replenishment in the surface layer during former spring tides, given less vertical mixing during neap tide, biomass increases in the upper layers (Fig. 10g). Resuspension effects resulting in increased turbidity at lower layers are visible from neap to spring but do not significantly change turbidity in the surface layers (Fig. 10f). Surface turbidity changes are consequences of increased NPP (Fig. 10f).

Observation-based estimates of spring–neap impacts on NPP given by Richardson et al. (2000) found that increased nitrate fluxes by tidal pumping contributed to NPP with 4–$6 \, \text{gC m}^{-2}$ for one spring–neap cycle at the northern edge of Dogger Bank, mainly due to increased production in the subsurface layer. By upscaling these results to the entire stratified season, considering six to eight spring–neap cycles, Richardson et al. (2000) proposed that the additional NPP contribution by the spring–neap cycle was in the range of 24–$48 \, \text{gC m}^{-2}$ for the whole stratified season. We resampled the simulated NPP along the same transect as sampled by Richardson et al. (2000) and for the same time period (29 July–4 August 1997). We extended the time period to 26 July–8 August 1997 to cover a full spring–neap cycle and found our simulated response of NPP to tidal forcing (the tidal scenario – non-tidal scenario) is $3.03 \, \text{gC m}^{-2}$ for one spring–neap cycle (Fig. A5a). These values are slightly below the lower edge of Richardson's estimates (4–$6 \, \text{gC m}^{-2}$). However, simulated frontal locations are not always conformed to the observed fronts due to unresolved subscale processes, which remain unconsidered in a $10 \, \text{km} \times 10 \, \text{km}$ model resolution and coarse atmospheric forcing (NCEP/NCAR reanalysis). When we resampled the NPP along the fronts in our simulation, which is at a distance of a few grid points further south from the fronts in Richardson et al. (2000) (Fig. A5a), we found that the simulated change in NPP ($5.99 \, \text{gC m}^{-2}$ for one spring–neap cycle) reaches the upper level of estimates based on observations (Fig. A5a, Table A1). When we compare the NPP response throughout the whole stratified season (simulated as

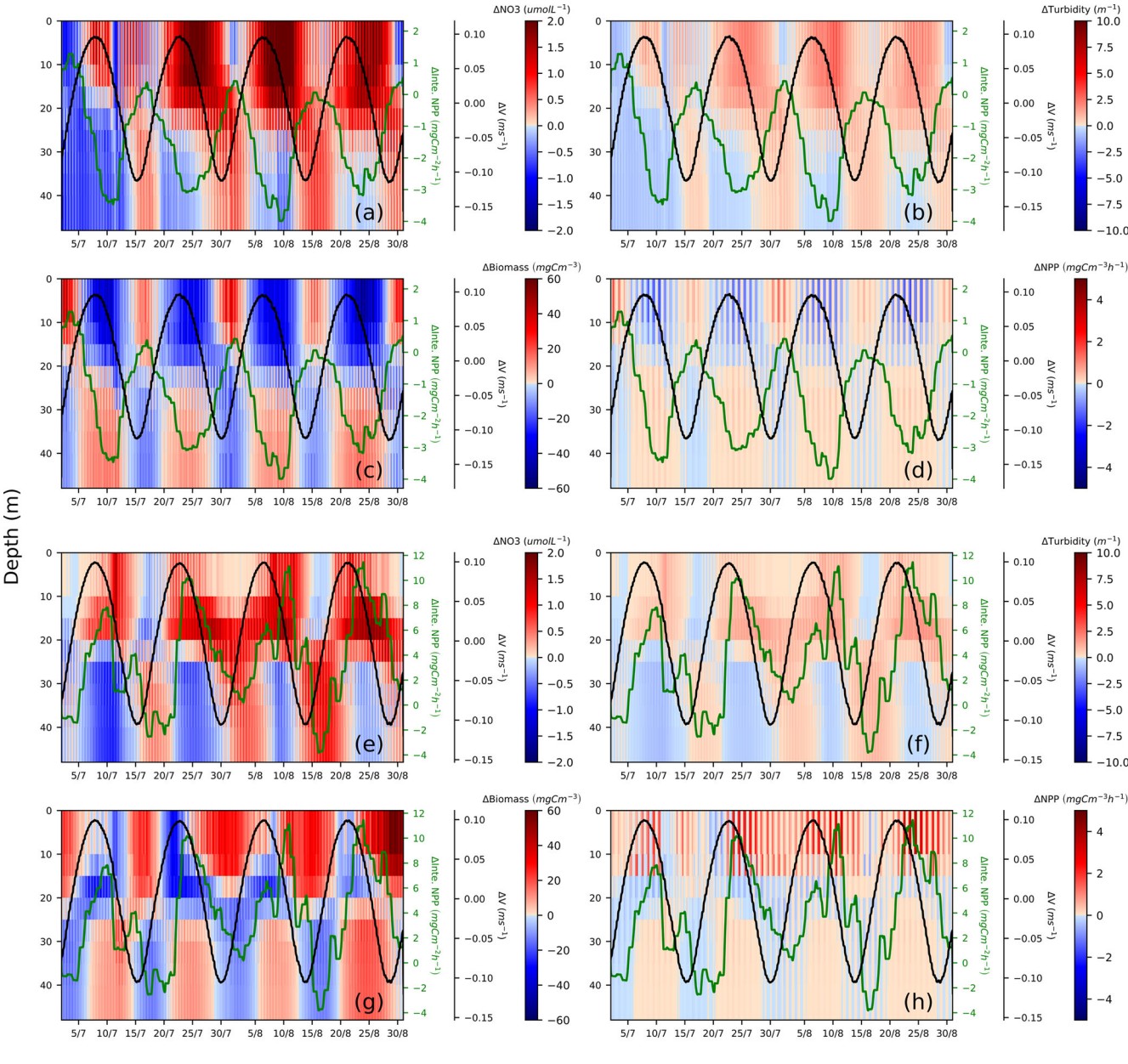

**Figure 10.** Spring–neap cycle impact on nitrate **(a, e)**, turbidity **(b, f)**, phytoplankton biomass **(c, g)** and primary production **(d, h)**. Differences between the two scenarios (tidal scenario $(M_2 + S_2) - M_2$ tidal scenario) are presented for two characteristic points, i.e., EH: **(a)**–**(d)**, WDB: **(e)**–**(h)**. To show the periodical fluctuation of currents and NPP and relate these fluctuations to changes in nitrate, turbidity, biomass and NPP, the differences in the depth-averaged velocity amplitude (black) and depth-integrated NPP (green) are presented in each subplot; both time series underwent smoothing with a 24 h running mean.

$15\,\mathrm{gC\,m^{-2}}$) we find this to be lower than Richardson's upscaling estimation ($24$–$48\,\mathrm{gC\,m^{-2}}$ for the whole stratified season). The reason for this discrepancy is a oversimplified upscaling procedure used by Richardson et al. (2000), neglecting the sensitivity to seasonality. Conditions measured over a few days between July and August (Richardson et al., 2000) are not representative of the whole strat-

ified season. In contrast to Richardson et al. (2000)'s conclusion that spring–neap cycle played the major role in fueling NPP, our study indicates further, that the semidiurnal tide plays the major role in pumping up nutrients and sustaining the NPP but not the spring–neap cycle as hypothesized by Richardson et al. (2000). Our estimate of, on average, $0.14\,\mathrm{gC\,m^{-2}}$ of NPP promoted by the spring–

neap tide during one tidal cycle (Fig. A5b) is considerably lower than that supported by the standard tidal forcing ($M_2 + S_2$) (5.99 gC m$^{-2}$) (Fig. A5a), and hence spring–neap tidal pumping contributes only little to the increase in NPP. Based on our simulation, tidal pumping sustaining subsurface NPP mainly occurs in July and August, with an average value of approximately 3 gC m$^{-2}$ month$^{-1}$ in frontal areas around Dogger Bank. This is close to the estimate of Richardsons et al. (2000). In other weakly stratified months, the value is no more than 1 gC m$^{-2}$ month$^{-1}$ or even negative (data not shown).

Sharples (2008) investigated a similar question for the Celtic Sea with model simulations. He found that the NPP varied up to 70 % with the spring–neap tidal cycle. We could not confirm such high tidal impacts on NPP for the North Sea; our estimates of the response of NPP to the spring–neap tidal cycle are only up to approximately 10 % of the tidal impact ($M_2 + S_2$) on NPP (Figs. 2 and 9). One explanation for this discrepancy is the higher spring–neap tidal current amplitude in the Celtic Sea compared to the North Sea, which may result in a stronger response of NPP to the spring–neap cycle. However, it is also possible that the simpler model setup used by Sharples, such as neglect of advection, a constant grazing rate and neglected impacts on resuspension and shading by DOM and detritus, resulted in higher NPP sensitivity to tidal forcing in their simulation.

As discussed in Sect. 3.2, tidal forcing not only impacts the magnitude of NPP but also spring bloom phenology. It is reasonable to assume that spring–neap tidal forcing also modulates the development of the spring bloom. To understand the impact of the spring–neap phase on the biomass build-up during the spring bloom, which typically occurs over one or several spring–neap cycles, we related the time periods with the maximum difference in NPP between the tidal scenario and the $M_2$ scenario to the spring–neap cycle phase (Fig. 11) at the SN (Spring–Neap) site (see Fig. 9). The SN site is located in the tidally energetic northwestern North Sea, where the development of the spring bloom often benefits from thermal stratification (Rodhe, 1998) but is sensitive to episodic "noise" added by wind forcing (Waniek, 2003). During the development of the spring bloom, in the difference between NPP time series (tidal scenario $-$ $M_2$ scenario), an increase in NPP often occurs in neap phases, whereas NPP is often decreased in spring phases. This indicates that the development of the spring bloom benefits from the neap phase but is interrupted or dampened during the spring tide (Fig. 11). A similar phenomenon has been explored and confirmed by Sharples et al. (2006) at a site south of the SN site. As suggested by Sharples et al. (2006), the onset time of the spring bloom is shifted by the spring–neap tidal cycle because the onset or intensity of stratification is strengthened during neap tides when the vertical mixing is dampened.

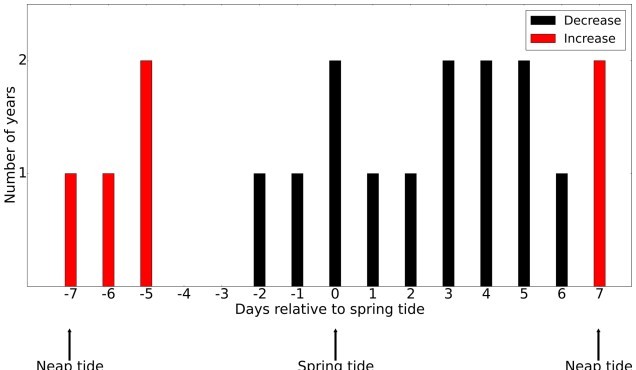

**Figure 11.** The occurrence of an increase (red) or decrease (black) in the NPP difference (tidal scenario ($M_2 + S_2$) $- M_2$ tidal scenario) relative to the nearest spring tide (spring and neap phase indicated). The development of the spring bloom period is defined as the time when NPP increases from 12.5 % to 87.5 % of the maximum NPP prior to the major peak of the spring bloom. TS3

## 4 Summary and conclusions

A model-based sensitivity experiment with varied tidal forcing was performed to evaluate tidal impacts on NPP, considering the major bottom-up controlling processes, including the tidal mixing of nutrients, organic matter and plankton biomass and tidal resuspension of suspended matter. The responses to tides in the North Sea differ regionally and depend on the local hydrodynamic characteristics. In permanently mixed areas in the southern part of the North Sea, light availability is the major limiting factor. The enhanced tidal resuspension and mixing of suspended matter into the surface layers deteriorate light conditions in the upper layers for phytoplankton growth and thus hinder primary production. In contrast, in frontal areas and seasonally stratified areas in the SNS where stratification is susceptible to tidal mixing, nutrient replenishment due to tidal forcing sustains NPP in summer and thus contributes a significant increase in NPP in both the surface layer and within the pycnocline. In the NNS, which is characterized by relatively weak tidal forcing and deep bathymetry, the bottom and upper mixed layers are well separated, and the influence of tidal forcing on NPP is limited.

However, the quantitative estimates provided here are model and parameterization specific. Dominant biochemical processes are generally well represented in simplified NPZD-type models, and the ECOSMO model used here is applicable for resolving ecosystem dynamics at seasonal to decadal timescales when forced by realistic boundary conditions (Daewel and Schrum, 2017). However, parameterization and unconsidered processes, such as the role of macrobenthos in the system, internal waves at the shelf break and coastal light attenuation due to inorganic suspended matter, and simplified physiological processes could potentially modulate or change the model's sensitivity to tidal forcing.

Please note the remarks at the end of the manuscript.

Studies identifying the contribution of these processes to tidal impacts on primary production are needed; thus far, we can only speculate on potential impacts.

Macrobenthic grazing likely changes the biochemical cycling and turbidity in the water column, subsequently changing the sensitivity of NPP to tidal forcing. In shallower waters, high near-bottom concentrations of suspended organic matter are susceptible to mixing into the euphotic zone, and increasing light attenuation leads to decreasing production (see Fig. 7c; cf. Sect. 3.1). Macrobenthic biomass, specifically from filter feeders, might significantly reduce resuspension and near-bottom suspended matter concentrations, thereby increasing the proportion of organic matter that remains in the food web (Prins et al., 1996). From observations, we know that macrobenthos show a distinct spatial pattern following principle production patterns in the North Sea with higher biomass in the shallow SNS (Heip et al., 1992). Therefore, we can expect an increase in NPP sensitivity to tidal forcing due to macrobenthos activity.

Furthermore, we hypothesize that the positive response of NPP to tidal forcing in the NNS was underestimated by our simulation due to the implementation of identical boundary conditions for all scenarios. We neglected the influence of tidal-generated internal waves on nutrient conditions. Tidal-generated internal waves are initiated at the shelf edge and enhance turbulent mixing at the shelf break and on the shelf (Heathershaw et al., 1987; Loder et al., 1992; New and Da Silva, 2002; Sharples et al., 2001). As internal tides break at the shelf edge, energy dissipates mainly at the shelf break and other bathymetric features, which causes vertical mixing that drives vertical nutrient fluxes and sustains phytoplankton growth (Holligan et al., 1985; Pingree et al., 1981; Sharples et al., 2007). Therefore, internal tidal waves will likely lead to mixing and increase nutrient pulses onto the shelf, consequently supporting NPP. In our setup, the average impact of tidal-generated internal waves on nutrient concentrations was considered with the climatological boundary conditions (Conkright et al., 2002), and differences among the simulated scenarios were not considered.

Another source of uncertainty in our model stems from the neglecting feedbacks related to inorganic material, which influences underwater light conditions, especially in shallow areas. Seasonal differences in yellow substance concentrations coincide with freshwater input (Schaub and Gieskes, 1991; Warnock et al., 1999). There are two main sources of SPM plumes in the North Sea. One source lies at the southern British coast and originates from local discharges (Humber–Wash and Thames rivers), coastal erosion and influx from the English Channel (Eisma, 2009). The other major source of SPM originates from the large continental rivers and diffusive sources entering the North Sea from the European continental coast, particularly off the Belgian coast and the Wadden Sea (van Alphen, 1990; Postma, 1981). Waves and currents are the controlling factors of the dispersion, resuspension and deposition processes of SPM (Holt and James,

1999). In winter, the two SPM plumes expand further offshore due to intensified mixing and both SPM plume deposits in both the Skagerrak and Norwegian channels. We have evaluated the potential impacts of inorganic SPM on our findings (Appendix C) and found that the general results regarding the tidal impacts on NPP remain largely insensitive to consideration of inorganic SPM and its seasonality. The reason lies in the spatial and temporal distribution of the SPM in the North Sea. During summer, SPM concentrations are low (Fig. C1) especially in stratified conditions and upper water layers (Capuzzo et al., 2013; Dobrynin et al., 2010). Only in shallow areas with permanent mixing, SPM concentrations are high (Van Raaphorst et al., 1998). This is critical for our analysis since most differences in NPP actually occur in summer stratified conditions. A simulation study (Tian et al., 2009) in the German Bight found that implementing SPM is only critical at the onset of bloom, given reasonable parameterization, similar bloom amplitude was achieved in scenarios including or omitting SPM. Furthermore, measurements suggested that in the central North Sea, the water body itself triggers most of the attenuation (Jones et al., 1998). SPM is more relevant to attenuation in nearshore areas due to cliff erosion and river input (Eisma, 2009). The relevance to turbidity of fluvial SPM is confined to river mouths because SPM deposits quickly (Pleskachevsky et al., 2011; Siegel et al., 2009). Organic suspended matter (which is considered in the model) accounts for a high fraction of the total suspended matter (TSM) in most areas in the southern North Sea except for the very nearshore areas (Schartau et al., 2018). The areas where inorganic suspended matter dominates are in the negatively responding regions of our analysis (Fig. 2c). The distribution of inorganic suspended matter is influenced by many factors, such as transportation with residual currents, aggregation with organic matter, type of benthic sediments and so on. Clearly, interaction processes as mentioned above cannot be resolved by implementing a climatological SPM field. Thus, the numerical experiment presented here is considered to be a first step towards understanding the role of SPM for tidal impacts, and further studies specifically focusing on shallow coastal areas would require reasonable boundary conditions for inorganic matter from benthic sediments and river inputs as well as a more reasonable representation of biophysical interactions related to inorganic matter. However, this is beyond the scope of the current study and should be emphasized more thoroughly in future work.

The major tidal impacts on NPP are via vertical mixing. Given the small horizontal gradient of both nutrients and biomass and weak tidal residuals of no more than a few centimeters per second (Prandle, 1984), the impacts of horizontal advection are negligible. To investigate the influence of advection on the concentration of nutrients and phytoplankton biomass in our study, we estimated the net horizontal transport between grid cells. In most parts of the study domain, we found that the contribution of mean tidal advection does not exceed 5% (not shown here). Exceptions oc-

cur in the Skagerrak Channel, where relatively high residual currents drive water exchange between the North Sea and Baltic Sea (Brettschneider, 1967), and in the EC close to the model boundary, where relatively high current speeds caused by atmospheric forcing and topography emerge irregularly, mainly in spring and winter. However, this result is not true for smaller horizontal and temporal resolutions.

Since the North Sea can in general be considered as bottom-up controlled (Daewel et al., 2014; Heath, 2005), using a lower trophic level model for investigating tidal impacts on NPP is a valid approach. Although situations with clear top-down control on zooplankton have been observed (Munk and Nielsen, 1994), these events occurred highly restricted in time and space and assumed to be only of minor relevance for the general processes described in this paper. In previous studies, which addressed similar scientific questions, constant grazing rates (Sharples, 2008) or grazing loss proportional to phytoplankton biomass (Cloern, 1991) were prescribed in the simulations. In this study, we utilize a lower trophic level NPZD-type model, only considering lower trophic level dynamics up to zooplankton, which is simulated as a state variable considering feeding preference, growth, excretion and mortality. Fish predation is only implicitly considered as part of the zooplankton mortality rate. Simulations with ECOSMO E2E (an updated version of the ECOSMO model) including functional groups for fish and macrobenthos revealed that temporal and spatial variations in zooplankton mortality due to fish predation are determined by the specific hydrodynamics of the North Sea (Daewel et al., 2018). Repeating a similar study with an NPZD-Fish model would be interesting; however, it is beyond the scope of our study.

Given the importance of tidal forcing for NPP, especially in frontal areas, which are known to be biological hotspots (Belkin et al., 2009), tidal impacts on higher trophic levels than those studied here merit further consideration and investigation in the future. Regarding the growth of macrobenthos, tidal stirring influences the sinking and resuspension of organic matter and thus influences food quality and bioturbation (Foshtomi et al., 2015; Zhang and Wirtz, 2017). Tidal forcing in frontal areas not only provides enough prey for fish larvae due to nutrient enrichment and higher NPP but also influences convergence zones, which are typical places for fish spawning and nursing (Bakun, 2006). Further investigations based on a combination of observations and multiprocess coupled simulations could enable a better understanding of the impacts of tidal forcing on ecosystem processes and their variability. Long-term tidal variations, such as the 18.61-year nodal cycle or the 8.85-year lunar perigee cycle, merit particular consideration.

**Data availability.** Simulated data sets in this study are currently not publicly accessible but are available on request.

## Appendix A

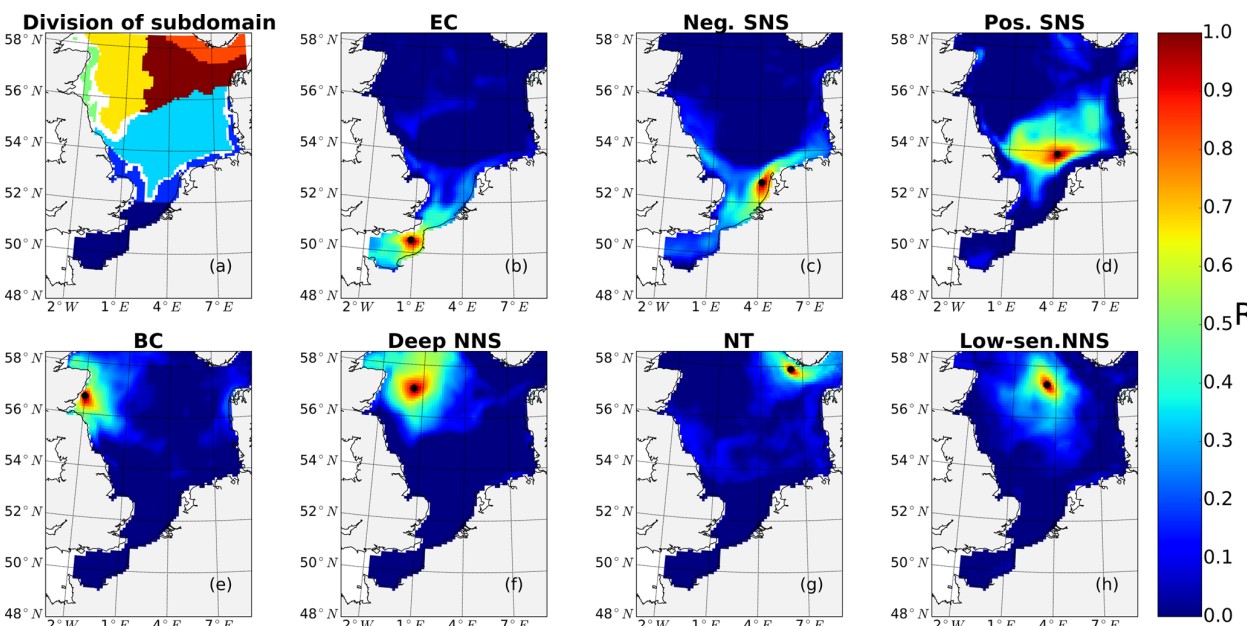

**Figure A1.** Subdomain divisions (**a**), correlation coefficient of NPP variations at the most representative grid cell (black dot) and NPP variations in the surrounding grid cells for the English Channel (**b**), negatively responding area in the southern North Sea (**c**), positively responding area in the southern North Sea (**d**), the eastern British coast (**e**), the deeper part of northern North Sea (**f**), the Norwegian Trench (**g**) and the low-sensitivity area in the northern North Sea (**h**).

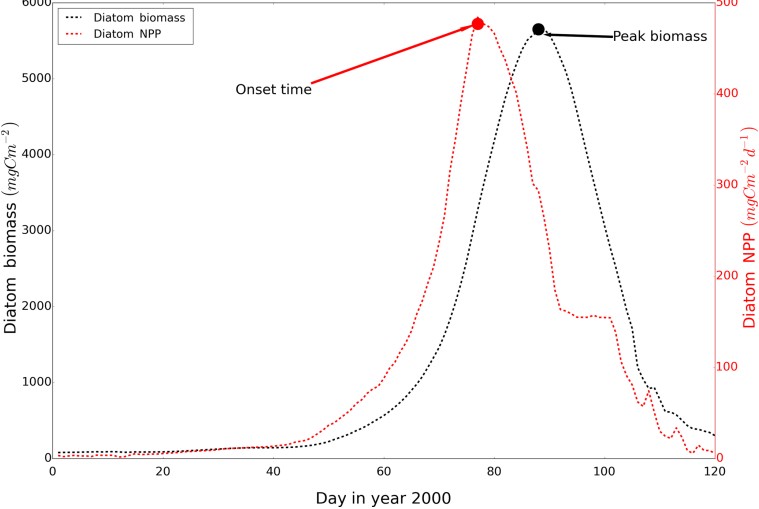

**Figure A2.** The definition of onset time of the spring bloom. The dashed black line is the time series of diatom biomass and the dashed red line is the time series of NPP. Both time series have undergone a 15 d running average. The black arrow depicts the time when the spring bloom reaches its maximum biomass. The red arrow depicts the time when the NPP reaches its maximum prior to biomass peak, which is defined as the onset time of the spring bloom. The time series is extracted from a grid cell (61° N, 4.2° E) from the ECOSMO simulation, for the year 2000.

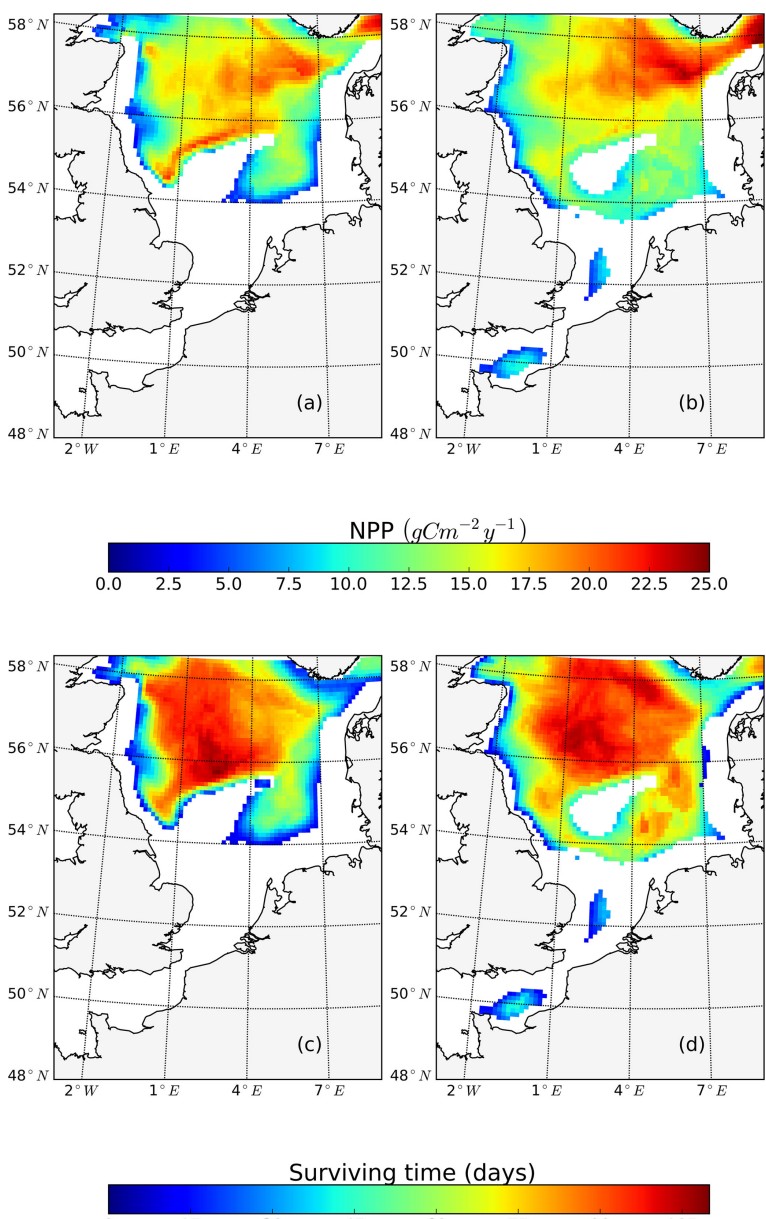

**Figure A3.** Annual mean NPP contributed by SBM in the tidal (**a**) and non-tidal scenarios (**b**). Survival time of the SBM for the tidal (**c**) and non-tidal (**d**) scenarios.

**Table A1.** NPP contributed by tidal forcing in the transect where Richardson et al. conducted their observation (northern edge of DB) and in the transect where the most pronounced front is located in our simulation (frontal transect).

| | Difference tide $(M_2 + S_2)$ − tide $(M_2)$ $(gC\,m^{-2}$ per spring–neap cycle) | Difference tide $(M_2 + S_2)$ − no tide $(gC\,m^{-2}$ per spring–neap cycle) |
|---|---|---|
| Northern edge of DB | 0.11 | 3.03 |
| Frontal transect | 0.14 | 5.99 |

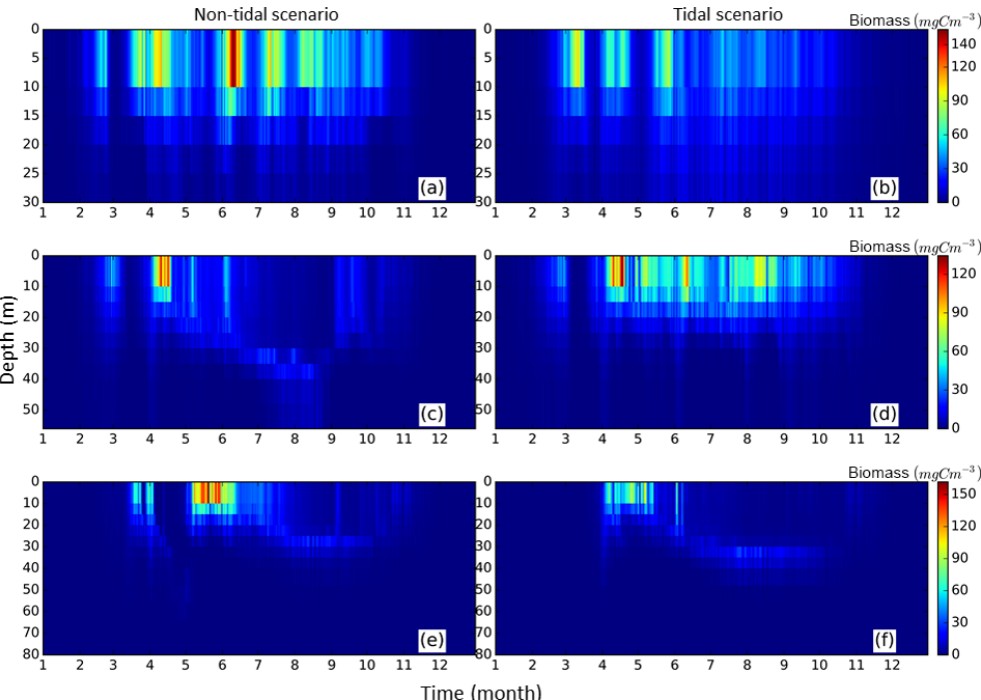

**Figure A4.** Annual mean (1990–2015) time series of vertical biomass profiles in the tidal (**b, d, f**) and non-tidal (**a, c, e**) scenarios at representative grid cells for the neg. SNS, the negatively responding southern North Sea (**a, b**), pos. SNS, the positively responding southern North Sea (**c, d**) and for the deep NNS, the deeper northern North Sea (**e, f**).

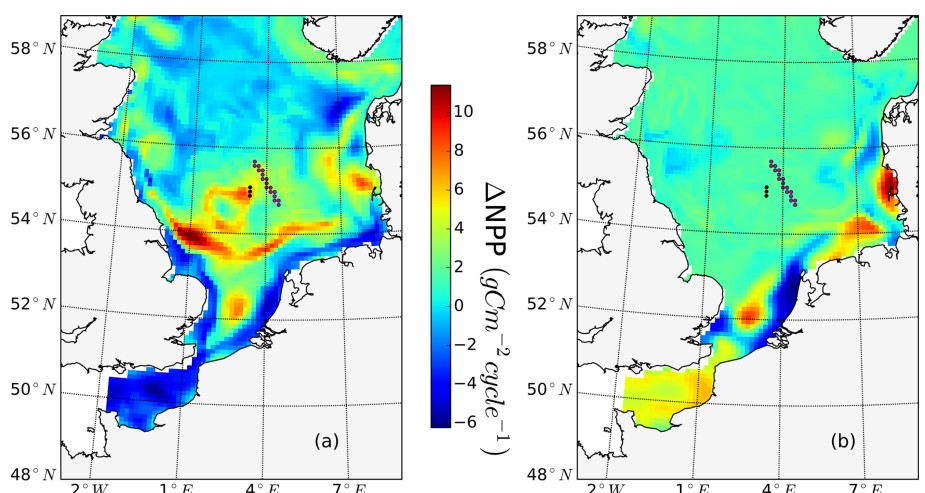

**Figure A5.** Vertically integrated NPP contributed by tide ($M_2 + S_2$) (**a**) and spring–neap tide (**b**) for one spring–neap cycle (26 July–8 August 1997) during the observational period studied by Richardson et al. (2000). Magenta dots depict the location of the transect which Richardson et al. (2000) has analyzed. Black dots depict the exact location of fronts in our simulation.

**Appendix B**

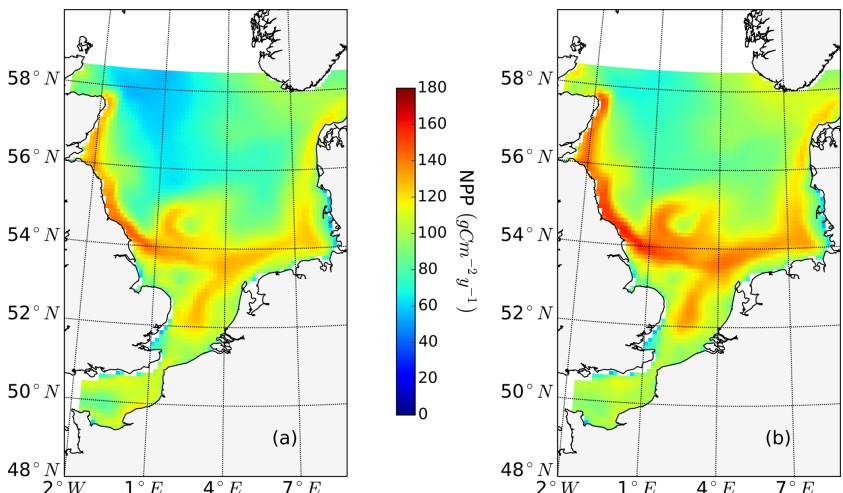

**Figure B1.** Mean annual net primary production for the analyzed period (1990–2015) simulated with the model configuration used by Daewel and Schrum (2013) **(a)** and the setup used in this study **(b)**.

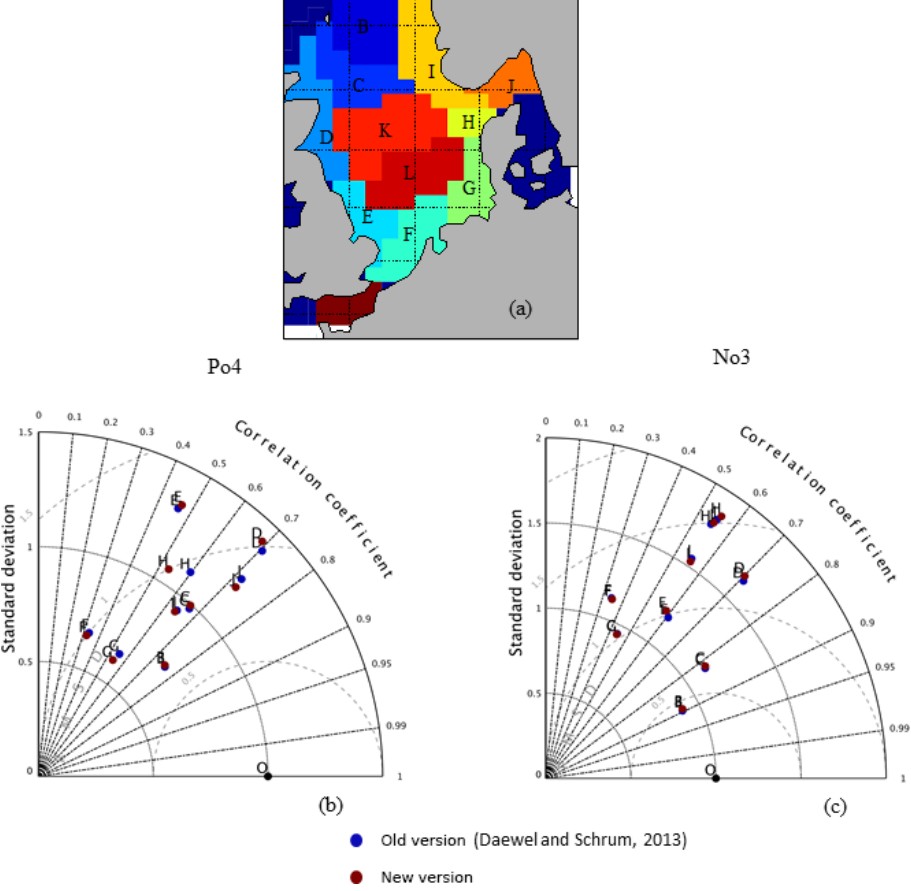

**Figure B2.** Taylor diagram for surface (above 20 m) nutrient validation (model vs. ICES data) in different areas of the North Sea for phosphate **(b)** and nitrogen **(c)**. Area separation is given in panel **(a)**.

## Appendix C

To estimate the impact of SPM on the underwater light climate and primary production dynamics in the simulation, we implemented a climatological SPM field for the North Sea (available with daily resolution and 31 vertical layers) for our simulation. This SPM field was derived from a statistical regression model which considers tidal currents, salinity and water depth (Heath et al., 2002). The SPM field is able to resolve the spatial distribution pattern and seasonal cycling of SPM concentration in the North Sea (Fig. C1) and has been applied in many hydrodynamical–biogeochemical coupled models (Große et al., 2016; Kerimoglu et al., 2017).

Taking the parameterization scheme proposed by Tian et al. (2009), we parameterize shading effects due to SPM as

$$Kd_{\mathrm{spm}} = k_{\mathrm{spm}} \cdot \sqrt{\mathrm{SPM}}. \tag{C1}$$

The $k_{\mathrm{spm}}$ was set as $0.02\,\mathrm{m^2\,g^{-1}}$. We added the contribution of SPM to the light shading scheme as described in the paper (Eq. 1). We decreased the background attenuation coefficient $k_{\mathrm{w1}}$ (0.03) to $0.025\,\mathrm{m^{-1}}$ ($k_{\mathrm{w2}}$) and use the following parameterization for light attenuation:

$$Kd_1 = k_{\mathrm{w2}} + k_{\mathrm{p}} \cdot P + k_{\mathrm{DOM}} \cdot \mathrm{DOM} + k_{\mathrm{Det}} \cdot \mathrm{Det} \\ + k_{\mathrm{spm}} \cdot \sqrt{\mathrm{SPM}}. \tag{C2}$$

We implemented the new light shading scheme (Eq. C2) and evaluated the difference in NPP contributed by tide, by comparing the annual mean NPP in tidal and non-tidal scenarios using Eq. (C2) (Fig. C2). The general pattern remains largely insensitive to consideration of spatial and seasonal variations in SPM. The positive and negative responding areas hold the same distribution pattern, but the NPP's increasing amplitude with tidal forcing in frontal areas decreases slightly when SPM is explicitly considered. This is because the elevated NPP fueled by pumped-up nutrients is partly offset by increased shading effects due to SPM. However, the sensitivity to SPM is minor and does not affect the general results of our study.

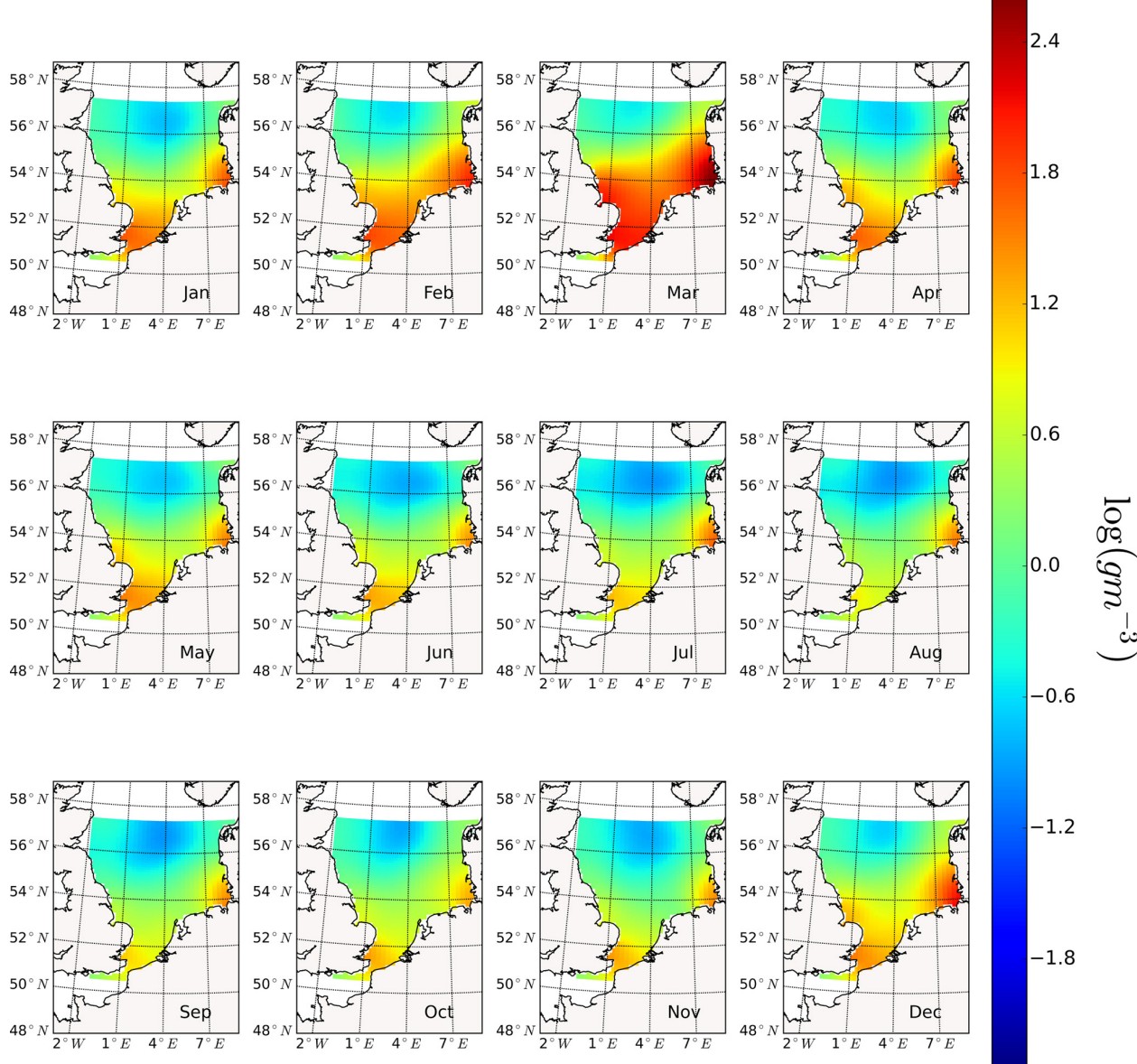

**Figure C1.** Monthly mean of inorganic SPM concentration in the first layer (upper 5 m).

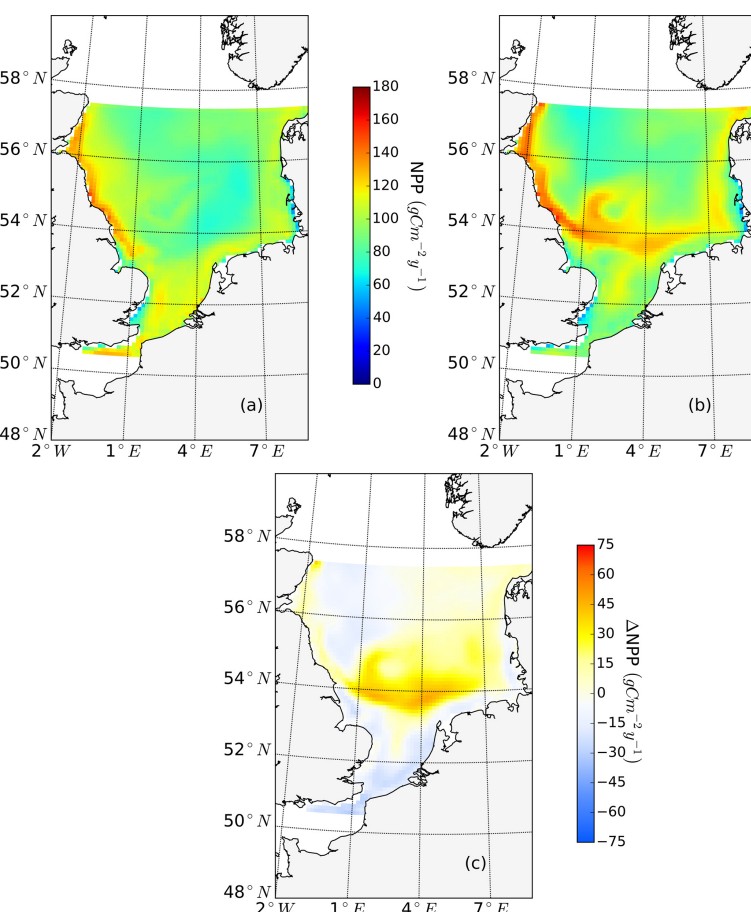

**Figure C2.** Mean annual net primary production for the analyzed period (1990–2015) of the non-tidal scenario **(a)** and tidal scenario **(b)**, both with SPM field implemented. The difference in the mean annual NPP of both scenarios is in panel **(c)**. The spatial coverage is smaller than original simulation domain since the SPM field data are available from 50.5 to 57.5° N.

**Author contributions.** ECOSMO code was maintained and provided by UD TS5. New added parameterization for SPM field, scenario runs, data analysis and most graphical presentations were conducted by CZ under supervision of UD. Story building and text writing were conducted by CZ under supervision of UD and CS. Conception and overall supervision of the paper were done by CS. CE2

**Competing interests.** The authors declare that they have no conflict of interest.

**Acknowledgements.** This work is funded by the Chinese Scholarship Council (no. 201406140121). We would like to thank Sonja M. van Leeuwen, Marie Maar and Johanes Pätsch for kindly sharing data with us. We appreciate the help provided by Richard Hofmeister related to technical issues in figure plotting.

The article processing charges for this open-access publication were covered by a Research Centre of the Helmholtz Association.

**Review statement.** This paper was edited by Yun Liu and reviewed by Thomas Pohlmann and Chuan-Yuan Hsu.

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

## Remarks from the language copy-editor

CE1    Please confirm the change.

CE2    Please confirm the minor change.

## Remarks from the typesetter

TS1    Please confirm change throughout.

TS2    Please note that it is our house standard to write out "Section" only at the beginning of a sentence.

TS3    Could you please try zooming in or printing this page out? The figure seems to be complete. If the problem persists, please let me know.

TS4    Please note that this change will have to be approved by the editor. Please provide a short statement explaining this correction that can be forwarded by us to the editor. Thank you very much in advance.

TS5    Please note that according to our house standards we do not use academic titles.

TS6    Please provide publisher.