# Peer review of "Tidal impacts on primary production in the North Sea"

_Earth System Dynamics, 2018_

## Referee Comment (RC1) · Anonymous Referee #1 · 30 Oct 2018

General Remarks: The paper investigates the impact of tides on the distribution of phytoplankton and other ecosystem parameters. For this purpose, three scenarios were conducted with the well-established ecosystem model ECOSMO, i.e. a non-tidal scenario, a tidal scenario (M2 and S2 considered) and a M2 scenario (only M2 considered). In order to also demonstrate the existing inter-annual variability, the model simulation was conducted for a 25 years' period (1990 to 2015). The overall impression is that the paper is written very carefully and in a clear and concise way. In particular, the scenario tests were chosen nicely to convincingly illustrate the impact of tides on the North Sea ecosystem dynamics. However, I have one principle concern, which should be considered by the authors before publication can be envisaged. As stated by the authors, the impact of tides is mainly governed by two counteracting processes.

[Figure]

On the one hand, an increased turbulence can transport more nutrients from deeper layers into the upper euphotic zone, which increases primary production. On the other hand, tides will increase the suspended particulate matter (SPM) concentration in the water column, which due to the shading effect will decrease primary production. Unfortunately, in the current paper version, the latter process is not described adequately. The present limitation to only organic SPM is not acceptable to describe the dynamics of suspended matter realistically enough, in particular in the southern North Sea. There are numerous publications, in particular those, which make use of remote sensing data, showing an important contribution of inorganic SPM input by rivers but also due to cliff erosion along the English east coast. From satellite observations it can be derived that these contributions dominate the SPM dynamics in the southern North Sea (see e.g. Pleskachevsky et al. JPO 2010). I would strongly suspect that ignoring the effect of inorganic SPM leads to an overestimation of the tidally induced SPM effect, since the background SPM concentration is too small, if only inorganic SPM is considered. Altogether, the entire presentation of the SPM dynamics is not acceptable as it is in the current version. As stated, compared to their reference paper Daewel & Schrum (2013), the implementation of the SPM dynamics was significantly modified. If this is the case, a thorough validation of this strongly modified scheme is indispensable, in particular since SPM dominates the light attenuation. The latter belongs to one of the two major processes affected by the tidal forcing, and therefore should be in the focus of the current paper. No figures or even numbers regarding the SPM concentration are provided. Therefore, it is not even possible to judge whether they are in the correct range at least. A general criticism of minor importance is the missing predation by fish and higher trophic levels. This deficit is only mentioned in the conclusions. However, a more serious discussion of this aspect would definitely be appropriate, in particular since it was noted in line 129 that the predator - prey interaction is considered, which at the first glance is even misleading. In summary, after the effect of inorganic SPM will have been considered appropriately, the manuscript will be suitable for publication in Earth System Dynamics after a moderate revision, accounting for the few minor

remarks given below.

Detailed Comments: Line 129: It is not clear that the mentioned predator - prey interaction just concerns zooplankton and phytoplankton, whereas fish is not considered. This must be clarified. Line 138: The term "southern coast" should be specified more clearly. Line 280: The sentence is not clear. How can the "energy" of tidal currents interact with the atmospheric forcing? Moreover, it is not clear whether this specific interaction process is considered in this study. I guess so, but however, this should be stated. Line 540: Obviously, the difference to observation is larger than one order of magnitude. The arguments, which are presented to defend this inconsistency are not fully convincing to explain such a very large discrepancy. In particular, the argument given at line 547 that observations over a few days between July and August cannot be compared with seasonally averaged model data is not acceptable. It should be easy to extract the actual observation period from a 25 years' model results data set. By this means a direct comparison could easily be performed. Line 552: Please correct to "close to".

---

## Referee Comment (RC2) · Anonymous Referee #2 · 9 Dec 2018

This paper attempts to describe the relation of tidal dynamics with NPP on the distribution of phytoplankton and other factors related to NPP growth rate over the North Sea through numerical simulations. For this purpose, three scenarios were applied through the physical-biogeochemical coupled model ECOSMO, which is non-tidal, tidal (M2 + S2), and M2 tidal scenario, respectively, from 1990 to 2015. Overall speaking, this paper is cautiously studied and is well described concisely on the impacts of the tide dynamics. As described, however, tides influence the biogeochemical cycle in two different ways. One of them is enhancing the vertical biomass mixing into the euphotic zone so that sustains the primary production. Another way is by vertical mixing diluting the biomass so that reducing the productivity. The latter process seems to not be well discussed in the current version of the manuscript. Other than this, the current

manuscript only has few remarks I want to address below:

1. A point of view as a modeler, I am somehow confused with the meaning of the spatial resolution 6' x 10' (line 91) because a) no unit associated with, and b) it is not the comment way we are using.

2. Line 138, "southern coast" is not clear to understand

3. Between line 197-204, authors divided the North Sea into three subdomains by tidal forcing, and then further separate it with positive net primary production and negative one. After, authors separate the southern North Sea into EC and outside of EC, separate the northern North Sea into NT and the deeper area. Those of sentences are not clear until figure 4 is mentioned. Please make it clear.

4. Line 219 – 226 also makes confuse to me. It looks like the authors want to further discuss the described impact on/before line 218. The descriptions, however, didn't well expound. For example, the definition of stratification is defined by the vertical seawater temperature difference reaching to 0.5 deg-C; however, why 0.5 deg-C is using here didn't explain. Also, how the averaged MLD can be used to measure the depth of stratification needs to be stated.

5. Line 229 – 237 and line 280 needs to well describe.

---

## Author Comment (AC1) · 5 Jan 2019

We appreciate the thoughtful and helpful comments on our manuscript. We reply below. Some contents were added in the supplementary materials, which are referred to in our answer.

1) The present limitation to only organic SPM is not acceptable to describe the dynamics of suspended matter realistically enough, in particular in the southern North Sea.

Response: We agree that inorganic SPM has the potential to reduce primary production, specifically in tidally influenced shallow water. We considered this effect through a slightly elevated background attenuation (line120 in the submitted version, Eq.S2 in

the supplementary materials). We further assumed that the spatial variability in SPM can be neglected for the sensitivity study performed here. However, the reviewer is right and our assumption needs to be verified. We agree that the impact of inorganic matter on light attenuation merits further analysis and have performed further sensitivity experiment to verify our assumption, which we discuss below. We will include a respective discussion as outlined below in a revised version of the manuscript.

To address the uncertainties related to SPM, we tested the effect of spatial-temporal varied inorganic SPM on our findings while performing an additional numerical sensitivity experiment. Here we implemented a climatological SPM filed (daily resolution, with 31 vertical layers) (Fig.S1) and added the SPM's contribution explicitly in the light attenuation scheme. Details of the SPM data set and implementation are given in the supplementary material. By running the tidal/non-tidal scenarios again using the new light attenuation scheme, we evaluated the impact of tides on NPP firstly by comparing annual mean NPP between tidal scenario and non-tidal scenario. We found the most significant change appearing in the frontal area where the tidal induced NPP elevation was decreased by about 10 gCmˆ(-2) yˆ(-1) (Fig.S2) compared to the original version (Fig. 2c), which indicates that in the frontal area SPM's impact dampens the promotion process on NPP by nutrients pumping. However, the positive and negative responding pattern as identified by the original simulations remain consistent even after considering spatial and seasonal variations in SPM (Fig.S2). This confirms that the general mechanism discussed in the manuscript and our conclusions regarding the former parameterizations remain valid.

Many earlier published studies support our assumption and the conclusion of the additional sensitivity experiment. First, with regard to the seasonality, SPM concentration and contribution to turbidity are low during summer (see also Fig.S1), (Capuzzo et al., 2013; Dobrynin et al., 2010), which is critical in our analysis since most differences in NPP actually occur and accumulate in summer. Measurements suggested that in the central North Sea, the water body itself triggers most of the attenuation; in Oyster

grounds, attenuation has been controlled to a large extend by CDOM and phytoplankton; SPM in the surface layer decreases after the onset of stratification (Jones et al., 1998). The SPM is more relevant to attenuation in nearshore area due to cliff erosion and river input. Astoreca et al. (2009) suggested that CDOM is mainly derived from local autochthonous rather than terrestrial source in offshore waters (salinity>34). The relevance to turbidity of fluvial SPM is confined to river mouths because SPM deposits quickly (Pleskachevsky et al., 2011; Siegel et al., 2009). In spring, simulation study in the German Bight found that implementing SPM is only critical at the onset of bloom, given reasonable parameterization, similar bloom amplitude was achieved in both scenarios including or omitting SPM (Tian et al., 2009). Horizontally, organic suspended matter shares a high fraction of total suspended matter (TSM) in most areas in the southern North Sea except in very near shore areas. The area where inorganic matter dominates reaches $8.5°E$ in stormy season (autumn) and are confined further inshore in summer (Schartau et al., 2018). The inorganic suspended matter dominating areas are in the negative responding regions based on our simulation results (Fig.2c). Considering enhanced resuspension and further attenuation caused by tidal forcing, the NPP in the near shore area would also respond negatively.

The distribution of inorganic suspended matter is influenced by many factors, such as transportation with residual currents, aggregation with organic matter, type of benthic sediments and so on. Clearly, interaction processes as mentioned above cannot be resolved by implementing a climatological SPM field. Thus, the numerical experiment presented can are a first step towards understanding tidal impacts, and future studies are suggested, given reasonable boundary conditions of inorganic matter from benthic sediments and river inputs as well as a more reasonable representation of bio-physical interactions related to inorganic matter. However, this is beyond the scope of the current study and should be emphasized more thoroughly in future work.

(2)As stated, compared to their reference paper Daewel & Schrum (2013), the implementation of the SPM dynamics was significantly modified. If this is the case, a thorough validation of this strongly modified scheme is indispensable, in particular since SPM dominates the light attenuation.

Response: Yes, we agree that due to the modification of the parameterization, an assessment of the changed model performance is necessary. New parameterizations of sedimentary respiration and light shading have been implemented in the new version (following Nissen, 2014)). Therefore, we will add a discussion of changes in mean primary production pattern and we will repeat the validation exercise proposed by Daewel and Schrum, (2013)) focusing on surface nutrient concentration, and compared results from with and without the new parameterizations. We found only small changes in production pattern from the new parameterization introduced (Fig S4). Frontal production is slightly enhanced and production increased slightly in deeper stable stratified waters and decreased weakly near the coast. The rigid validation of nutrient surface concentrations also revealed only small changes (Fig.S3). Here we found that the performance of the model in the North Sea region is rather stable and changes only marginally. The validation will be included in the revised manuscript.

(3) General criticism of minor importance is the missing predation by fish and higher trophic levels. This deficit is only mentioned in the conclusions. However, a more serious discussion of this aspect would definitely be appropriate, in particular since it was noted in line 129 that the predator - prey interaction is considered, which at the first glance is even misleading.

Response: We agree that the definition of predator-prey interaction is misleading and will define this more appropriately as predator-prey interaction at lower trophic levels between the considered functional groups of zooplankton and phytoplankton in the model.

Since the North Sea can in general be considered as bottom-up controlled (Daewel et al., 2014; Heath, 2005), using a lower trophic level model for investigating tidal impacts on NPP is a valid approach. Although situations with clear top-down control on zoo-

plankton has been observed (Munk and Nielsen, 1994), these events occurred highly restricted in time and space and assumed to be only of minor relevance for the general processes described in this manuscript. However, we will include a more thorough discussion about the relevance of fish predation in the discussion. In previous studies, which addressed similar scientific question, constant grazing rate (Sharples, 2008) or grazing loss being proportional to phytoplankton biomass (Cloern, 1991) were prescribed in their simulations. In this study, we utilize a lower trophic level NPZD-type model only considering lower trophic level dynamics up to zooplankton, which is simulated as a state variable considering feeding preference, growth, excretion and mortality. Fish predation is only implicitly considered as part of the zooplankton mortality rate. Simulations with ECOSMO E2E (an updated version of the ECOSMO model) including functional groups for fish and macrobenthos revealed that temporal and spatial variations in zooplankton mortality due to fish predation are determined by the specific hydrodynamics of the North Sea (Daewel et al., 2018). Repeating a similar study with an NPZD-Fish model would be interesting, however, beyond the scope of our study.

(4) Line 138: The term "southern coast" should be specified more clearly.

Response: Yes, we agree that this term is ambiguous. In the updated version, we change it as "European continental coast".

(5) Line 280: The sentence is not clear. How can the "energy" of tidal currents interact with the atmospheric forcing? Moreover, it is not clear whether this specific interaction process is considered in this study. I guess so, but however, this should be stated.

Response: We agree that it is necessary to change the way this is expressed and a similar comment was added by reviewer 2. To address the comments of both reviewers, line 280 was changed to "In addition to tidal forcing, atmospheric forcing and bathymetry modulates stratification (Van Leeuwen et al., 2015) and productivity pattern (Daewel and Schrum, 2017); consequently tidal impacts on stratification and hence primary production are subject to spatial-temporal variability."

(6) Line 540: Obviously, the difference to observation is larger than one order of magnitude. The arguments, which are presented to defend this inconsistency are not fully convincing to explain such a very large discrepancy. In particular, the argument given at line 547 that observations over a few days between July and August cannot be compared with seasonally averaged model data is not acceptable. It should be easy to extract the actual observation period from a 25 years' model results data set.

Response: This seem to be a misunderstanding. Here we explore a discussion and conclusion by Richardson et al. (2000). The upscaling of the short-term observation to seasonal pattern was initially proposed by Richardson et al. (2000). They upscaled their measured NPP (4-6 gCmˆ(-2)per spring neap cycle during 29 July to 4 August, 1997, in their publication) to the whole stratified season (May to October) which contains 6-8 times of spring neap cycle as they assumed. Based on this simple upscaling, they suggested NPP contributed by the spring neap cycle of about 24-48 gCmˆ(-2) yˆ(-1). We believe that this upscaling is too simplistic and discussed the mean local impacts based on our simulations to provide dynamically consistent estimates. Making use of our simulations, we have analyzed monthly variability regarding to the NPP contributed by tides. As we have pointed out in our manuscript (line 547-553), the strongest contribution by tides to NPP is in June, July and August; in other seasons, the contributions are weaker or even negative. In the supplementary materials, we also provided monthly mean contribution to NPP by tides to prove that the monthly variability is considerable and has to be considered (Fig. S5). To avoid misunderstanding, we replace the sentence in line 536: "The subsurface NPP attributed by nitrate fluxes driven by spring-neap tides by Richardson et al. were 4-6 gCmˆ(-2) for one spring-neap cycle; considering 6-8 times of spring neap cycle during the whole stratified season per year, they did an upscaling and proposed that the additional NPP contributed by the spring neap cycle was in the range of 24-48 gCmˆ(-2) yˆ(-1). "

We added a direct comparison between our results with the observations made by Richardson. For the comparison between our simulation and Richardson's observa-

tion, we have extracted NPP at the exact location where they did their measurements for the same period (29/07-04/08 1997 in Richardson et al., 2000. We extended it to 26/07/1997-08/08/1997 to cover a full spring-neap cycle). In the previous version, we only used the NPP generated in the subsurface layer to make comparison, since they stated that the NPP was mainly generated in the subsurface layer (Richardson et al., 2000). However, in their study, they used the integrated oxygen surplus in the whole water column to estimate NPP. We think it would make more sense to use integrated NPP in our simulation to compare with Richardson et al.'s results. It is true that our simulated changes in NPP is smaller than the observed changes at the same location (the magenta transect, Fig.S6, Table S1). However, we found substantial small scale variability in the response to tidal forcing at the order of a gird cell (Fig. S6) and only at a distance of 3 grid points further south where the front locates in our simulation (the black transect, Fig.6), the modelled tidal contribution (M2+S2) reaches the level with the observed value (Table. S1). We think that the discrepancy stem from uncertainties introduced by unresolved sub-scale processes, which remain unconsidered in a 10km x 10 km model resolution and coarse scale atmospheric forcing (NCEP/NCAR reanalysis); intensity of simulated fronts is likely influenced. Keeping the uncertainties in estimating the exact location of a front in mind when comparing to point-observations, we think that the overall response of the model is rather consistent with observations and can be used to assess the overall tidal vs the spring-neap tidal impact to update Richardson et al. estimates of tidal impacts on primary production and to conclude improved seasonal mean estimates.

We will improve the discussion in the revised manuscript to clarify our point.

We agree that inorganic SPM has the potential to reduce primary production, specifically in tidally influenced shallow water. We considered this effect through a slightly elevated background attenuation (line120 in the submitted version, Eq.S2 in the supplementary materials). We further assumed that the spatial variability in SPM can be neglected for the sensitivity study performed here. However, the reviewer is right and our assumption needs to be verified. We agree that the impact of inorganic matter on light attenuation merits further analysis and have performed further sensitivity experiment to verify our assumption, which we discuss below. We will include a respective discussion as outlined below in a revised version of the manuscript.

To address the uncertainties related to SPM, we tested the effect of spatial-temporal varied inorganic SPM on our findings while performing an additional numerical sensitivity experiment. Here we implemented a climatological SPM filed (daily resolution, with 31 vertical layers) (Fig.S1) and added the SPM's contribution explicitly in the light attenuation scheme. Details of the SPM data set and implementation are given in the supplementary material. By running the tidal/non-tidal scenarios again using the new light attenuation scheme, we evaluated the impact of tides on NPP firstly by comparing annual mean NPP between tidal scenario and non-tidal scenario. We found the most significant change appearing in the frontal area where the tidal induced NPP elevation was decreased by about 10 $gCm^{-2}y^{-1}$ (Fig.S2) compared to the original version (Fig. 2c), which indicates that in the frontal area SPM's impact dampens the promotion process on NPP by nutrients pumping. However, the positive and negative responding pattern as identified by the original simulations remain consistent even after considering spatial and seasonal variations in SPM (Fig.S2). This confirms that the general mechanism discussed in the manuscript and our conclusions regarding the former parameterizations remain valid.

Many earlier published studies support our assumption and the conclusion of the additional sensitivity experiment. First, with regard to the seasonality, SPM concentration and contribution to turbidity are low during summer (see also Fig.S1), (Capuzzo et al., 2013; Dobrynin et al., 2010), which is critical in our analysis since most differences in NPP actually occur and accumulate in summer. Measurements suggested that in the central North Sea, the water body itself triggers most of the attenuation; in Oyster grounds, attenuation has been controlled to a large extend by CDOM and phytoplankton; SPM in the surface layer decreases after the onset of stratification (Jones et al., 1998). The SPM is more relevant to attenuation in nearshore area due to cliff erosion and river input. Astoreca et al. (2009) suggested that CDOM is mainly derived from local autochthonous rather than terrestrial source in offshore waters (salinity>34). The relevance to turbidity of fluvial SPM is confined to river mouths because SPM deposits quickly (Pleskachevsky et al., 2011; Siegel et al., 2009). In spring, simulation study in the German Bight found that implementing SPM is only critical at the onset of bloom, given reasonable parameterization, similar bloom amplitude was achieved in both scenarios including or omitting

SPM (Tian et al., 2009). Horizontally, organic suspended matter shares a high fraction of total suspended matter (TSM) in most areas in the southern North Sea except in very near shore areas. The area where inorganic matter dominates reaches $8.5°E$ in stormy season (autumn) and are confined further inshore in summer (Schartau et al., 2018). The inorganic suspended matter dominating areas are in the negative responding regions based on our simulation results (Fig.2c). Considering enhanced resuspension and further attenuation caused by tidal forcing, the NPP in the near shore area would also respond negatively.

The distribution of inorganic suspended matter is influenced by many factors, such as transportation with residual currents, aggregation with organic matter, type of benthic sediments and so on. Clearly, interaction processes as mentioned above cannot be resolved by implementing a climatological SPM field. Thus, the numerical experiment presented can are a first step towards understanding tidal impacts, and future studies are suggested, given reasonable boundary conditions of inorganic matter from benthic sediments and river inputs as well as a more reasonable representation of bio-physical interactions related to inorganic matter. However, this is beyond the scope of the current study and should be emphasized more thoroughly in future work.

(2)As stated, compared to their reference paper Daewel & Schrum (2013), the implementation of the SPM dynamics was significantly modified. If this is the case, a thorough validation of this strongly modified scheme is indispensable, in particular since SPM dominates the light attenuation.

Yes, we agree that due to the modification of the parameterization, an assessment of the changed model performance is necessary. New parameterizations of sedimentary respiration and light shading have been implemented in the new version (following Nissen, 2014)). Therefore, we will add a discussion of changes in mean primary production pattern and we will repeat the validation exercise proposed by Daewel and Schrum, (2013)) focusing on surface nutrient concentration, and compared results from with and without the new parameterizations. We found only small changes in production pattern from the new parameterization introduced (Fig S4). Frontal production is slightly enhanced and production increased slightly in deeper stable stratified waters and decreased weakly near the coast. The rigid validation of nutrient surface concentrations also revealed only small changes (Fig.S3). Here we found that the performance of the model in the North Sea region is rather stable and changes only marginally. The validation will be included in the revised manuscript.

(3) General criticism of minor importance is the missing predation by fish and higher trophic levels. This deficit is only mentioned in the conclusions. However, a more serious discussion of this aspect would definitely be appropriate, in particular since it was noted in line 129 that the predator - prey interaction is considered, which at the first glance is even misleading.

We agree that the definition of predator-prey interaction is misleading and will define this more appropriately as predator-prey interaction at lower trophic levels between the considered functional groups of zooplankton and phytoplankton in the model.

Since the North Sea can in general be considered as bottom-up controlled (Daewel et al., 2014; Heath, 2005), using a lower trophic level model for investigating tidal impacts on NPP is a valid

approach. Although situations with clear top-down control on zooplankton has been observed (Munk and Nielsen, 1994), these events occurred highly restricted in time and space and assumed to be only of minor relevance for the general processes described in this manuscript. However, we will include a more thorough discussion about the relevance of fish predation in the discussion. In previous studies, which addressed similar scientific question, constant grazing rate (Sharples, 2008) or grazing loss being proportional to phytoplankton biomass (Cloern, 1991) were prescribed in their simulations. In this study, we utilize a lower trophic level NPZD-type model only considering lower trophic level dynamics up to zooplankton, which is simulated as a state variable considering feeding preference, growth, excretion and mortality. Fish predation is only implicitly considered as part of the zooplankton mortality rate. Simulations with ECOSMO E2E (an updated version of the ECOSMO model) including functional groups for fish and macrobenthos revealed that temporal and spatial variations in zooplankton mortality due to fish predation are determined by the specific hydrodynamics of the North Sea (Daewel et al., 2018). Repeating a similar study with an NPZD-Fish model would be interesting, however, beyond the scope of our study.

(4) Line 138: The term "southern coast" should be specified more clearly.

Yes, we agree that this term is ambiguous. In the updated version, we change it as "European continental coast".

(5) Line 280: The sentence is not clear. How can the "energy" of tidal currents interact with the atmospheric forcing? Moreover, it is not clear whether this specific interaction process is considered in this study. I guess so, but however, this should be stated.

We agree that it is necessary to change the way this is expressed and a similar comment was added by reviewer 2. To address the comments of both reviewers, line 280 was changed to "In addition to tidal forcing, atmospheric forcing and bathymetry modulates stratification (Van Leeuwen et al., 2015) and productivity pattern (Daewel and Schrum, 2017); consequently tidal impacts on stratification and hence primary production are subject to spatial-temporal variability."

(6) Line 540: Obviously, the difference to observation is larger than one order of magnitude. The arguments, which are presented to defend this inconsistency are not fully convincing to explain such a very large discrepancy. In particular, the argument given at line 547 that observations over a few days between July and August cannot be compared with seasonally averaged model data is not acceptable. It should be easy to extract the actual observation period from a 25 years' model results data set.

This seem to be a misunderstanding. Here we explore a discussion and conclusion by Richardson et al. (2000). The upscaling of the short-term observation to seasonal pattern was initially proposed by Richardson et al. (2000). They upscaled their measured NPP (4-6 $gCm^{-2}$ per spring neap cycle during 29 July to 4 August, 1997, in their publication) to the whole stratified season (May to October) which contains 6-8 times of spring neap cycle as they assumed. Based on this simple upscaling, they suggested NPP contributed by the spring neap cycle of about 24-48 $gCm^{-2}y^{-1}$. We believe that this upscaling is too simplistic and discussed the mean local impacts based on our simulations to provide dynamically consistent estimates. Making use of our simulations, we have analyzed monthly variability regarding to the NPP contributed by tides. As we have pointed out in our manuscript (line 547-553), the strongest contribution by tides to NPP

is in June, July and August; in other seasons, the contributions are weaker or even negative. In the supplementary materials, we also provided monthly mean contribution to NPP by tides to prove that the monthly variability is considerable and has to be considered (Fig. S5). To avoid misunderstanding, we replace the sentence in line 536: "The subsurface NPP attributed by nitrate fluxes driven by spring-neap tides by Richardson et al. were 4-6 $gCm^{-2}$ for one spring-neap cycle; considering 6-8 times of spring neap cycle during the whole stratified season per year, they did an upscaling and proposed that the additional NPP contributed by the spring neap cycle was in the range of 24-48 $gCm^{-2}y^{-1}$."

We added a direct comparison between our results with the observations made by Richardson. For the comparison between our simulation and Richardson's observation, we have extracted NPP at the exact location where they did their measurements for the same period (29/07-04/08 1997 in Richardson et al., 2000. We extended it to 26/07/1997-08/08/1997 to cover a full spring-neap cycle). In the previous version, we only used the NPP generated in the subsurface layer to make comparison, since they stated that the NPP was mainly generated in the subsurface layer (Richardson et al., 2000). However, in their study, they used the integrated oxygen surplus in the whole water column to estimate NPP. We think it would make more sense to use integrated NPP in our simulation to compare with Richardson et al.'s results. It is true that our simulated changes in NPP is smaller than the observed changes at the same location (the magenta transect, Fig.S6, Table S1). However, we found substantial small scale variability in the response to tidal forcing at the order of a gird cell (Fig. S6) and only at a distance of several grid points further south where the front exactly locates in our simulation (the black transect, Fig.6), the modelled tidal contribution (M2+S2) reaches the level with the observed value (Table. S1). We think that the discrepancy stem from uncertainties introduced by unresolved sub-scale processes, which remain unconsidered in a 10km x 10 km model resolution and coarse scale atmospheric forcing (NCEP/NCAR reanalysis); intensity of simulated fronts is likely influenced. Keeping the uncertainties in estimating the exact location of a front in mind when comparing to point-observations, we think that the overall response of the model is rather consistent with observations and can be used to assess the overall tidal vs the spring-neap tidal impact to update Richardson et al. estimates of tidal impacts on primary production and to conclude improved seasonal mean estimates.

We will improve the discussion in the revised manuscript to clarify our point.

Reference

Astoreca, R., Rousseau, V. and Lancelot, C.: Coloured dissolved organic matter (CDOM) in Southern North Sea waters: Optical characterization and possible origin, Estuar. Coast. Shelf Sci., 85(4), 633–640, doi:10.1016/j.ecss.2009.10.010, 2009.

[revised manuscript text omitted]

**Supplementary materials**

1) **Impact of SPM**. To estimate the impact of SPM on the under-water light climate in the simulation, we implemented a climatological SPM filed of the North Sea (with daily resolution and 31 vertical layers) in our simulation. This SPM filed was derived from statistical regression model which considers tidal currents, salinity and water depth (Heath et al., 2002). This SPM filed is able to resolve spatial distribution pattern and seasonal cycling of SPM concentration in the North Sea (Fig.S1). This SPM field has been applied in many hydrodynamic-biogeochemical coupling (Große et al., 2016; Kerimoglu et al., 2017).

Taking the parameterization scheme in (Tian et al., 2009), we evaluate shading effect due to SPM as:

$$Kd_{spm} = k_{spm} \cdot \sqrt{SPM} \qquad (S1)$$

The $k_{spm}$ was set as $0.02\ m^2 g^{-1}$. We added the contribution of SPM to the light shading scheme as described in the paper (Eq.1 in the submitted version). We decreased the background attenuation co-efficient $k_{w1}$, $0.03\ m^{-1}$, to $0.025\ m^{-1}$ ($k_{w2}$) and subsequently generated the new light shading scheme:

$$Kd_1 = k_{w2} + k_p \cdot P + k_{DOM} \cdot DOM + k_{Det} \cdot Det + k_{spm} \cdot \sqrt{SPM} \qquad (S2)$$

We implemented the new light shading scheme (Eq.S2) in the simulation and evaluated the difference in NPP contributed by tide, by comparing the annual mean NPP in tidal and non-tidal scenarios (Fig.S2). The general pattern remain. The positive and negative responding area hold the same distribution pattern, except for frontal areas where elevated NPP decreases slightly when SPM impact is explicitly considered. This is because the elevated NPP fueled by pumped up nutrients are partly offset by increased shading effects due to SPM. However, the changes are minor and do not affect the general sensitivity pattern.

2) **Subsurface NPP compared to observation**.
Table S1. NPP contributed by tidal forcing in the transect where Richardson did their observation (Northern edge of DB) and in the transect where fronts locate in our simulation (frontal transect), a few grid points away

|  | Difference Tide (M2 +S2) – Tide (M2) $(gCm^{-2}$ per spring-neap cycle) | Difference Tide (M2 +S2) - no-tide $(gCm^{-2}$ per spring-neap cycle) |
|---|---|---|
| Northern edge of DB | 0.11 | 3.03 |
| Frontal transect | 0.14 | 5.987 |

[Figure]

Figure S1. Monthly mean of inorganic SPM concentration in the first layer (upper 5 meters)

[Figure]

Figure S2. Mean annual net primary production for the *analysed period* (1990-2015) of the *non-tidal scenario* (a), and t*idal scenario* (b), both with SPM filed implemented. The difference in the mean annual NPP of both scenarios is in (c). The spatial coverage is smaller than original simulation domain since the SPM filed data is available from $50.5°N - 57.5°N$

[Figure]

Figure S3 Taylor diagram for surface (above 20 m) nutrient validation (model versus ICES data) in different areas of the North Sea for phosphate (b) and nitrogen (c). Area separation is given in (a).

[Figure]

Figure S4. Mean annual net primary production for the *analysed period* (1990-2015) of the former setup (Schrum and Daewel, 2013) (a ) and the setup in this study (b)

[Figure]

Figure S5. Monthly mean of NPP's response to tide (M2+S2). In the subsurface layer (a), with colorbar ranging from -6~6 $gCm^{-2}$; for the surface layer (b), the colorbar ranges from -16~16 $gCm^{-2}$

[Figure]

Figure S6. Vertically integrated NPP contributed by tide (M2+S2) (a) and spring-neap tidal cycle (b) for one spring neap cycle (26/07/1997-08/08/1997) as the same period when the measurements were taken in Richardson et al., 2000. Magenta dots depict the location of the transects which Richardson et al. (2000) has analyzed. Black dots depict the exact location of fronts in our simulation.

---

## Author Comment (AC2) · 5 Jan 2019

We appreciate the thoughtful and helpful comments on our manuscript. Please find our reply below. Figures are available in the supplementary materials

1) One of them is enhancing the vertical biomass mixing into the euphotic zone so that sustains the primary production. Another way is by vertical mixing diluting the biomass so that reducing the productivity. The latter process seems to not be well discussed in the current version of the manuscript.

Repponse: We agree. Both mechanisms (i.e. pumping up nutrients & dilution of biomass out of euphotic zone) play a significant role in modulating the tidal response of productivity. However, the impact of tidal mixing on phytoplankton biomass distribution resulting in a lower productivity has only been explored in the discussion of the negatively responding southern North Sea (line 385- 392 in the submitted version) and while discussing the impact of the spring neap cycle (see Fig.10g and Fig.10c and in the respective discussions line 516- 530). Our discussion in the submitted version is indeed a bit too brief and we will further expand the discussion in the revised version to hopefully clarify the involved processes.

In the following the mechanism is exemplarily further explored corresponding to Fig.10g and Fig.10c. During spring tide (depicted by high values in black line), increased mixing results in less phytoplankton biomass in the upper layer and more biomass in the lower layer. Enhanced vertical mixing during spring tide dilute the phytoplankton cells in the upper layer and redistribute them more evenly in the whole water column, thereby reducing productivity. For Fig.6, we also supplemented a plot below (please see it in the supplementary material), which shows the time evolution of vertical distribution of phytoplankton biomass at the representative point in neg.SNS (the same grid cell plotted in Fig.6 a,b,c,d). It shows that the tidal mixing smoothed the vertical gradient of phytoplankton biomass; in contrast, in the non-tidal scenario, the phytoplankton biomass tends shows higher concentrations in the upper layer.

We will further expand the discussion in the revised version to hopefully clarify the involved processes.

2) A point of view as a modeler, I am somehow confused with the meaning of the spatial resolution 6' x 10' (line 91) because a) no unit associated with, and b) it is not the comment way we are using.

Response: The model uses a spherical coordinate. To make this more clear we change the sentence into "The model was formulated on a staggered Arakawa-C grid using spherical coordinates, with a spatial resolution of 6' in latitude and 10' in longitude.

3) Line 138, "southern coast" is not clear to understand

Response: Yes, we agree that this term is ambiguous. In the updated version, we change it as 'European continental coast

4) Between line 197-204, authors divided the North Sea into three subdomains by tidal forcing, and then further separate it with positive net primary production and negative one. After, authors separate the southern North Sea into EC and outside of EC, separate the northern North Sea into NT and the deeper area. Those of sentences are not clear until figure 4 is mentioned. Please make it clear.

Response: Yes, we agree that this is somewhat unclear. In a revised version of the manuscript we will make the logic of sub-area division more clear and reconstruct the paragraph: "The pre-division of the area into subdomains is based on a combination of geographic location, bathymetry and the local responses of NPP to tidal forcing (increase, decrease). First, SNS and NNS were divided by the 65 m isobath. In the SNS, areas with positive and negative NPP response to tides were separated. The negatively responding area in the SNS was further geographically divided into the English Channel (EC, south of 52°N) and an area along the continental coast (neg.SNS). In the NNS, the area of the Norwegian Trench (NT) was separated, which was characterized by a water depth deeper than 200 m. The remaining region of the NNS was further divided based on the response of NPP to tidal forcing. The area along the eastern British coast (BC) showing elevated NPP in response to tides was separated from the negative responding area in the middle of NNS (deep NNS). In the east of the NNS, an area with mild increase of NPP was identified (low-sen. NNS)."

(5). Line 219 – 226 also makes confuse to me. It looks like the authors want to further discuss the described impact on/before line 218. The descriptions, however, didn't well expound. For example, the definition of stratification is defined by the vertical seawater temperature difference reaching to 0.5 deg-C; however, why 0.5 deg-C is using here didn't explain. Also, how the averaged MLD can be used to measure the depth of stratification needs to be stated.

Response: In coastal and shelf seas, stable stratification with lighter water above heavier water emerges as a consequence of an increase in buoyancy from surface heating and/or freshwater input counteracting mixing processes (from tides, waves, winds). Once stratification establishes, the water column form a layer, which separated the upper surface mixed layer from the deep water and acts as a barrier that dampens vertical mixing and exchange of materials. Except for regions of fresh water influence (ROFI) the dominant reason for stratification is surface heating, which has a strong seasonal cycle in the North Sea resulting in seasonal stratification pattern (Schrum et al., 2003). That is why we use the temperature difference to identify the depth of the surface mixed layer (MLD) to quantify the stratification. The difference of temperature reaching 0.5 âĎĆ between surface and layers below is a criterion we have chosen to identify the onset of stratification and mixed layer depth. The same method has also been used in many other studies (Gong et al., 2014; Karl and Lukas, 1996; Lefèvre et al., 1994; Richardson et al., 2002; Sharples et al., 2006). This will be clarified in a revised version of the manuscript.

(6). Line 229 – 237 and line 280 needs to well describe.

Response: Lines 229-237 will be rewritten as : " The onset of the spring bloom, the establishment of stratification and sufficient light conditions were estimated in days of the year for each grid point for each simulation year; subsequently the percentage of years with advanced and/or delayed responses to tidal forcing were estimated for each grid point. The increase/decrease of winter zooplankton and peak amplitude of spring bloom after applying tidal forcing, were also recorded for each grid cell for each year. Subsequently, we obtained the spatial pattern of percentage of years with 1) higher amplitude of the spring bloom, 2) later onset of the spring bloom, 3) later onset of stratification, 4) deeper mixed layer depth, 5) later occurrence of sufficient light conditions for building phytoplankton biomass, and 6) higher concentration of winter zooplankton in response to tidal forcing (tidal scenario vs. non-tidal scenario)."

Line 280 will be rewritten as: "In addition to tidal forcing, atmospheric forcing and

bathymetry modulates stratification (Van Leeuwen et al., 2015) and productivity pattern (Daewel and Schrum, 2017); consequently tidal impacts on stratification and hence primary production are subject to spatial-temporal variability."

Reference

Daewel, U. and Schrum, C.: Low-frequency variability in North Sea and Baltic Sea identified through simulations with the 3-D coupled physical-biogeochemical model ECOSMO, Earth Syst. Dyn., 8(3), 801–815, doi:10.5194/esd-8-801-2017, 2017.

Gong, X., Shi, J. and Gao, H.: Modeling seasonal variations of subsurface chlorophyll maximum in South China Sea, J. Ocean Univ. China, 13(4), 561–571, doi:10.1007/s11802-014-2060-4, 2014.

Karl, D. M. and Lukas, R.: The Hawaii Ocean Time-series (HOT) program: Background, rationale and field implementation, Deep Sea Res. Part II Top. Stud. Oceanogr., doi:10.1016/0967-0645(96)00005-7, 1996. Van Leeuwen, S., Tett, P., Mills, D. and Van Der Molen, J.: Stratified and nonstratified areas in the North Sea: Long-term variability and biological and policy implications, J. Geophys. Res. C Ocean., 120(7), 4670–4686, doi:10.1002/2014JC010485, 2015.

Lefèvre, D., Bentley, T. L., Robinson, C., Blight, S. P. and Williams, P. J. I. B.: The temperature response of gross and net community production and respiration in time-varying assemblages of temperate marine micro-plankton, J. Exp. Mar. Bio. Ecol., doi:10.1016/0022-0981(94)90005-1, 1994.

Richardson, A. J., Pfaff, M. C., Field, J. G., Silulwane, N. F. and Shillington, F. A.: Identifying characteristic chlorophyll a profiles in the coastal domain using an artificial neural network, J. Plankton Res., 24(12), 1289–1303, doi:10.1093/plankt/24.12.1289, 2002.

Schrum, C., Siegismund, F. and John, M. S.: Decadal variations in the stratification and circulation patterns of the North Sea. Are the 1990s unusual?, ICES Mar. Sci. Symp.,

2003.

Sharples, J., Ross, O. N., Scott, B. E., Greenstreet, S. P. R. and Fraser, H.: Inter-annual variability in the timing of stratification and the spring bloom in the North-western North Sea, Cont. Shelf Res., 26(6), 733–751, doi:10.1016/j.csr.2006.01.011, 2006.

Please also note the supplement to this comment:
https://www.earth-syst-dynam-discuss.net/esd-2018-74/esd-2018-74-AC2-supplement.pdf

---

## Author Response (AR1)

**Response letter for reviewer 1**

We appreciate the thoughtful and helpful comments on our manuscript. We reply below. Original text from reviewers in black color and our answers are in blue color. We also lay out corresponding line numbers where modifications has made, in green color.

1) The present limitation to only organic SPM is not acceptable to describe the dynamics of suspended matter realistically enough, in particular in the southern North Sea.

We agree that inorganic SPM has the potential to reduce primary production, specifically in tidally influenced shallow water. In this study, we considered this effect through a slightly elevated background attenuation (Eq.1). We further assumed that the spatial variability in SPM can be neglected for the sensitivity study performed here. However, the reviewer is right and our assumption needs to be verified. We agree that the impact of inorganic matter on light attenuation merits further analysis and have performed further sensitivity experiment to verify our assumption, which we discuss below.

To address the uncertainties related to SPM, we tested the effect of spatial-temporal varied inorganic SPM on our findings byperforming an additional numerical sensitivity experiment. Here we implemented a climatological SPM filed (daily resolution within in one year, with 31 vertical layers) (Fig.C1) and added the SPM's contribution explicitly in the light attenuation scheme. Details of the SPM data set and implementation are given in Appendix C.. By running the tidal/non-tidal scenarios again using the new light attenuation scheme, we evaluated the impact of tides on NPP firstly by comparing annual mean NPP between tidal scenario and non-tidal scenario. We found the most significant change appearing in the frontal area where the tidal induced NPP elevation was decreased by about 10 $gCm^{-2}y^{-1}$ (Fig.C2) compared to the original version (Fig. 2c), which indicates that in the frontal area SPM's impact dampens the promotion process on NPP by nutrients pumping. However, the positive and negative responding pattern as identified by the original simulations remain consistent even after considering spatial and seasonal variations in SPM (Fig.C2). This confirms that the general mechanism discussed in the manuscript and our conclusions regarding the former parameterizations remain valid.

Many earlier published studies support our assumption and the conclusion of the additional sensitivity experiment. First, with regard to the seasonality, SPM concentration and contribution to turbidity are low during summer (see also Fig.C1), (Capuzzo et al., 2013; Dobrynin et al., 2010), which is critical in our analysis since most differences in NPP actually occur and accumulate in summer. Measurements suggested that in the central North Sea, the water body itself triggers most of the attenuation; in Oyster grounds, attenuation has been controlled to a large extend by CDOM and phytoplankton; SPM in the surface layer decreases after the onset of stratification (Jones et al., 1998). The SPM is more relevant to attenuation in nearshore area due to cliff erosion and river input. Astoreca et al. (2009) suggested that CDOM is mainly derived from local autochthonous rather than terrestrial source in offshore waters (salinity>34). The relevance to turbidity of fluvial SPM is confined to river mouths because SPM deposits quickly (Pleskachevsky et al., 2011; Siegel et al., 2009). In spring, simulation study in the German Bight found that implementing SPM is only critical at the onset of bloom, given reasonable parameterization, similar bloom amplitude was achieved in both scenarios including or omitting SPM (Tian et al., 2009). Horizontally, organic suspended matter shares a high fraction of total suspended matter (TSM) in most areas in the southern North Sea except in very near shore areas. The area where inorganic matter dominates reaches $8.5°E$ in stormy season (autumn) and are confined further inshore in summer (Schartau et al., 2018). The inorganic suspended matter dominating areas are in the negative responding regions based on our simulation results (Fig.2c). Considering enhanced resuspension and further attenuation caused by tidal forcing, the NPP in the near shore area would also respond negatively.

The distribution of inorganic suspended matter is influenced by many factors, such as transportation with residual currents, aggregation with organic matter, type of benthic sediments and so on. Clearly, interaction processes as mentioned above cannot be resolved by implementing a climatological SPM field. Thus, the numerical experiment presented can are a first step towards understanding tidal impacts, and future studies are suggested, given reasonable boundary conditions of inorganic matter from benthic sediments and river inputs as well as a more reasonable representation of bio-physical interactions related to inorganic matter. However, this is beyond the scope of the current study and should be emphasized more thoroughly in future work.

In the manuscript, consideration of SPM was first mentioned in the description of model setup (line 134- 143). In the result part, we add a short description regarding the influence of SPM (Line 325). In the conclusion part, we further discussed the uncertainty (line 636-664). Further details of scenarios considering SPM were laid out in Appendix C.

 (2)As stated, compared to their reference paper Daewel & Schrum (2013), the implementation of the SPM dynamics was significantly modified. If this is the case, a thorough validation of this strongly modified scheme is indispensable, in particular since SPM dominates the light attenuation.

Yes, we agree that due to the modification of the parameterization, an assessment of the changed model performance is necessary. New parameterizations of sedimentary respiration and light shading have been implemented in the new version (following Nissen, 2014)). Therefore, we will add a discussion of changes in mean primary production pattern and we will repeat the validation exercise proposed by Daewel and Schrum, (2013) focusing on surface nutrient concentration, and compared results from with and without the new parameterizations. We found only small changes in production pattern from the new parameterization introduced (Fig.B1). Frontal production is slightly enhanced and production increased slightly in deeper stable stratified waters and decreased weakly near the coast. The rigid validation of nutrient surface concentrations also revealed only small changes (Fig.B2). Here we found that the performance of the model in the North Sea region is rather stable and changes only marginally.

The update in the manuscript

We added a description of the updated setup used in this study in the methods part (line 112-113). A short description of the comparison between two setups were also added (line 125-129). Details of the validation were in Appendix B.

 (3) General criticism of minor importance is the missing predation by fish and higher trophic levels. This deficit is only mentioned in the conclusions. However, a more serious discussion of this aspect would definitely be appropriate, in particular since it was noted in line 129 that the predator - prey interaction is considered, which at the first glance is even misleading.

We agree that the definition of predator-prey interaction is misleading and define this more appropriately as phytoplankton-zooplankton feedbacks at lower trophic levels between the considered functional groups of zooplankton and phytoplankton in the model.

85  Since the North Sea can in general be considered as bottom-up controlled (Daewel et al., 2014; Heath, 2005), using a lower trophic level model for investigating tidal impacts on NPP is a valid approach. Although situations with clear top-down control on zooplankton has been observed (Munk and Nielsen, 1994), these events occurred highly restricted in time and space and assumed to be only of minor relevance for the general processes described in this manuscript. However, we will include a more

90  thorough discussion about the relevance of fish predation in the discussion. In previous studies, which addressed similar scientific question, constant grazing rate (Sharples, 2008) or grazing loss being proportional to phytoplankton biomass (Cloern, 1991) were prescribed in their simulations. In this study, we utilize a lower trophic level NPZD-type model only considering lower trophic level dynamics up to zooplankton, which is simulated as a state variable considering feeding preference, growth,

95  excretion and mortality. Fish predation is only implicitly considered as part of the zooplankton mortality rate. Simulations with ECOSMO E2E (an updated version of the ECOSMO model) including functional groups for fish and macrobenthos revealed that temporal and spatial variations in zooplankton mortality due to fish predation are determined by the specific hydrodynamics of the North Sea (Daewel et al., 2018). Repeating a similar study with an NPZD-Fish model would be interesting, however, beyond the

100  scope of our study.

The update in the manuscript

We first gave a more appropriate definition as 'predator-prey feedbacks' in line 132. We also added a discussion about this issue in the conclusion part (line 673-684).

(4) Line 138: The term "southern coast" should be specified more clearly.

105  Yes, we agree that this term is ambiguous. In the updated version, we change it as "European continental coast".

The update in the manuscript

We changed this word in line 150.

(5) Line 280: The sentence is not clear. How can the "energy" of tidal currents interact with the atmospheric forcing? Moreover, it is not clear whether this specific interaction process is considered in this study. I guess so, but however, this should be stated.

We agree that it is necessary to change the way this is expressed and a similar comment was added by reviewer 2. To address the comments of both reviewers, line 280 was changed to "In addition to tidal forcing, atmospheric forcing and bathymetry modulates stratification (Van Leeuwen et al., 2015) and productivity pattern (Daewel and Schrum, 2017); consequently tidal impacts on stratification and hence primary production are subject to spatial-temporal variability."
Update in the manuscript

This update can be found in the line 295-298.

(6) Line 540: Obviously, the difference to observation is larger than one order of magnitude. The arguments, which are presented to defend this inconsistency are not fully convincing to explain such a very large discrepancy. In particular, the argument given at line 547 that observations over a few days between July and August cannot be compared with seasonally averaged model data is not acceptable. It should be easy to extract the actual observation period from a 25 years' model results data set.

This seem to be a misunderstanding. Here we explore a discussion and conclusion by Richardson et al. (2000). The upscaling of the short-term observation to seasonal pattern was initially proposed by Richardson et al. (2000). They upscaled their measured NPP (4-6 $gCm^{-2}$ per spring neap cycle during 29 July to 4 August, 1997, in their publication) to the whole stratified season (May to October) which contains 6-8 times of spring neap cycle as they assumed. Based on this simple upscaling, they suggested NPP contributed by the spring neap cycle of about 24-48 $gCm^{-2}$ for the whole stratified season. We believe that this upscaling is too simplistic and discussed the mean local impacts based on our simulations to provide dynamically consistent estimates. Making use of our simulations, we have analyzed monthly variability regarding to the NPP contributed by tides. As we have pointed out in our manuscript (line 575-577.), the strongest contribution by tides to NPP is in June, July and August; in other seasons, the contributions are weaker or even negative. For the whole stratified season, the tidal contribution to the NPP in the frontal area in our simulation is 15 $Cm^{-2}$.

We added a direct comparison between our results with the observations made by Richardson. For the comparison between our simulation and Richardson's observation, we have extracted NPP at the exact location where they did their measurements for the same period (29/07-04/08 1997 in Richardson et al., 2000. We extended it to 26/07/1997-08/08/1997 to cover a full spring-neap cycle). In the previous version, we only used the NPP generated in the subsurface layer to make comparison, since they stated that the NPP was mainly generated in the subsurface layer (Richardson et al., 2000). However, in their study, they used the integrated oxygen surplus in the whole water column to estimate NPP. We think it would make more sense to use integrated NPP in our simulation to compare with Richardson et al.'s results. It is true that our simulated changes in NPP (3.03 $gCm^{-2}$ for one spring neap cycle) is smaller than the observed changes at the same location (the magenta transect, Fig.A5, Table S1). However, we

found substantial small scale variability in the response to tidal forcing and only at a distance of several grid points further south where the front exactly locates in our simulation (the black transect, Fig.A5), the modelled tidal contribution ($M_2+S_2$) reaches the level (5.99 $gCm^{-2}$ per spring neap cycle ) with the observed value (Table. A1). We think that the discrepancy stem from uncertainties introduced by unresolved sub-scale processes, which remain unconsidered in a 10km x 10 km model resolution and coarse scale atmospheric forcing (NCEP/NCAR reanalysis); intensity of simulated fronts is likely influenced. Keeping the uncertainties in estimating the exact location of a front in mind, when comparing to point-observations, we think that the overall response of the model is rather consistent with observations and can be used to assess the overall tidal vs the spring-neap tidal impact to update Richardson et al. estimates of tidal impacts on primary production and to conclude improved seasonal mean estimates.

Update in the manuscript

We updated this discussion in the line 553-579.

305

310

315

320 The track version is made using the 'compare' function in WORD
software. A change in space could be marked out as total change of
the whole paragraph.

[revised manuscript text omitted]

---

## Referee Report (RR1)

Title: Tidal impacts on primary production in the North Sea

Comments:

The reviewed article has great revision updated. Here, only one point hopes the authors can explain it much clearly. As what addressed in the previous comments, I have asked that the authors should define/cite or should describe the definition of the depth of stratification (line 233-235). It is good that I received several references from authors to confirm the definition, saying that the temperature drops 0.5 deg-C from surface [Gong et al., 2014; Karl and Lukas, 1996; Richardson et al., 2002; Sharples et al., 2006].

Here, I summarized the use of those provided reference.

1. Gong et al. (2014), used this temperature difference to determine the depth of the mixed layer, but their paper focused on the south sea, which is around 8 deg-N ~ 25 deg-N

2. Karl and Lukas (1996), also use this criterion; however, they focused on the Hawaii island region, which is 25 deg-N around.

3. Richardson et al. (2002), used this method to calculate the MLD around Agulhas Bank [around 35 deg-S].

4. According to the Sharples et al., 2006, the difference of 0.5 deg-C temperature criterion is applied to determine the beginning period of onset stratification, on the difference of the temperature between surface and bottom water. Subsets is presented below.

"onset of the main period of stratification in Table 2. is defined as the first date after which the surface-bottom temperature differences exceeds 0.5 deg-C and is maintained for at least 3 days, preventing earlier short-term stratification events being identified as the beginning of the main summer stratification. ....."

In addition, in the response of the preview review, authors also provided a reference: Lefèvre, D., Bentley, T. L., Robinson, C., Blight, S. P. and Williams, P. J. l. B.: The temperature response of gross and net community production and respiration in time-varying assemblages of temperate marine micro-plankton, J. Exp. Mar. Bio. Ecol., doi:10.1016/0022-0981(94)90005-1, 1994.

I search the keyword by "Lefèvre" and "1994" in the article, I cannot find where this is cited. In my opinion, I believed that authors want to cite a paper that use the definition of mixed layer depth. If that is the case, I will use: Levitus 1982, Climatological atlas of the world ocean. This book addressed the difference of the MLD definition between temperature criterion (0.5 deg-C) and density criterion (0.125 kgm$^{-3}$).

Overall, I agreed with the most of the papers addressed above, using the 0.5 deg-C to determine the MLD in tropical/subtropical. The use of this criterion here is ok but acceptable. However, if like line 233-236 described, using this $\Delta T$ + time window (3 days) to determine the onset time of stratification, we might need to address here clearly. Because reference 4 is define the $\Delta T = 0.5$ deg-C differently. It should not make confuse for the future readers here by defining two different things with the same criterion with no clear explanation. Please classify this point clearly.

---

## Author Response (AR2)

Dear editor:

We appreciate the thoughtful and helpful comments on our manuscript.

For the citation relevant to the definition of stratification, we agree that the last citation (Sharples et al., 2006) should not be there. The method used by Sharples et al. is different from other 3 citations. The purpose for us to cite the paper is that Sharples et al. (2006) required that the onset of stratification should maintain at least 3 days to remove short-time oscillations. We also added this criterion for the onset time of stratification in our study. Thus, we should place the citation of (Sharples et al., 2006) after the sentence 'The time window (3 days) was chosen to filter out short-lived stratification variations and the day-night heating/cooling cycle.' in line 236.

For the definition of stratification onset time, we exactly use the criterion of 0.5 ℃' dropping of temperature from surface layer, as what those cited publications have done (Gong et al., 2014; Karl and Lukas, 1996; Richardson et al., 2002).

When we were preparing the revised manuscript (last version), we also found that the citation of Lefèvre 1994 is a mistake. There is no relevant content. That is why I deleted it in the revised manuscript.

reference

Gong, X., Shi, J. and Gao, H.: Modeling seasonal variations of subsurface chlorophyll maximum in South China Sea, J. Ocean Univ. China, 13(4), 561–571, doi:10.1007/s11802-014-2060-4, 2014.

Karl, D. M. and Lukas, R.: The Hawaii Ocean Time-series (HOT) program: Background, rationale and field implementation, Deep Sea Res. Part II Top. Stud. Oceanogr., doi:10.1016/0967-0645(96)00005-7, 1996.

Richardson, A. J., Pfaff, M. C., Field, J. G., Silulwane, N. F. and Shillington, F. A.: Identifying characteristic chlorophyll a profiles in the coastal domain using an artificial neural network, J. Plankton Res., 24(12), 1289–1303, doi:10.1093/plankt/24.12.1289, 2002.

Sharples, J., Ross, O. N., Scott, B. E., Greenstreet, S. P. R. and Fraser, H.: Inter-annual variability in the timing of stratification and the spring bloom in the North-western North Sea, Cont. Shelf Res., 26(6), 733–751, doi:10.1016/j.csr.2006.01.011, 2006.